SPECIAL ISSUE
CELL BIOLOGY OF THE NUCLEUS

# Changes in nuclear and actin mechanics from G1 to G2 affect nuclear integrity

Samantha Bunner[1],*, Katie Huang[1],*, Anish Shah[1],*, Schuyler Figueroa[1],*, Nick Lang[1], Catherine Chu[1], Nebiyat Eskndir[1], Mai Pho[1], Gianna Manning[1], Mindy Zheng[1], Lilian Fritz-Laylin[1,2,3], Katrina B. Velle[4], Joshua Marcus[5,6], James Orth[6] and Andrew D. Stephens[1,2,‡]

## ABSTRACT

The structural integrity of the nucleus is dependent on nuclear mechanical elements of chromatin and lamins resisting antagonistic actin cytoskeleton forces. Force imbalance results in nuclear blebbing, rupture and cellular dysfunction found in many human diseases. Here, we used the fluorescent ubiquitin cell cycle indicator (FUCCI) cells to determine how cell cycle changes affect the nucleus and actin force balance. Whereas nuclear blebs were present equally throughout interphase, nuclear blebs formed predominantly in G1 and then persisted into G2. Actin-based nuclear confinement and focal adhesion density was greater in G1 versus G2 cells. Removal of focal adhesions through treatment with an inhibitor resulted in decreased nuclear confinement and blebbing, supporting this as the underlying mechanism. Upon artificial confinement, G2 nuclei ruptured more than G1 nuclei. Single nucleus micromanipulation force measurements confirmed that G1 nuclei were stiffer than G2 nuclei in both the chromatin-based and lamin-based nuclear stiffness regimes. Decreased nuclear stiffness can be explained by loss of peripheral H3K9me3 from G1 to G2, recapitulated by H3K9me3 inhibition through treatment with chaetocin. Cell cycle-based changes in nuclear and actin mechanics impact nuclear integrity and shape.

KEY WORDS: Cell cycle, G1, G2, Nuclear bleb, Nuclear rupture, Actin confinement, Heterochromatin

## INTRODUCTION

The nucleus is the organelle that houses the genome and its essential functions. Nuclear mechanics, shape and integrity are the physical properties that ensure proper nuclear function. Loss of these physical properties is well-documented across the human disease spectrum including heart disease, aging and cancer (Stephens et al., 2019a; Kalukula et al., 2022). Nuclear blebs are a specific type of nuclear deformation of >1 μm in size hallmarked by decreased DNA density (Bunner et al., 2024; Chu et al., 2025; Pujadas Liwag et al., 2025). These highly curved nuclear blebs have a high propensity to result in nuclear rupture (Stephens et al., 2018; Xia et al., 2018). Loss of nuclear mechanics leading to nuclear blebbing and rupture causes nuclear dysfunction because of increased DNA damage (Denais et al., 2016; Irianto et al., 2016; Raab et al., 2016; Chen et al., 2018; Xia et al., 2018; Stephens et al., 2019b; Earle et al., 2020; Nader et al., 2021; Shah et al., 2021; Pho et al., 2023), changes in transcription (De Vos et al., 2011; Helfand et al., 2012; Berg et al., 2023) and perturbations to cell cycle control (Pfeifer et al., 2018), all believed to aid disease progression. It remains unknown how changes during interphase stages impact nuclear blebbing and rupture.

Changes during the cell cycle could affect nuclear and actin mechanics. For example, genome replication and the resulting doubling of genome content could cause alterations to chromatin, a major mechanical component of the nucleus. More specifically, chromatin histone modifications (Shimamoto et al., 2017; Stephens et al., 2017, 2018, 2019b; Hobson et al., 2020; Nava et al., 2020; Williams et al., 2024; Danielsson et al., 2022; Manning et al., 2025), H1 dynamics (Furusawa et al., 2015; Senigagliesi et al., 2019), chromatin linking proteins (Strom et al., 2021; Williams et al., 2024) and chromosome conformation interactions (Belaghzal et al., 2021) are all known contributors to nuclear mechanics and/or shape that might be altered during or after genome replication. DNA transcription activity has been shown to be essential for nuclear blebbing through a non-bulk mechanical property hypothesized to be due to chromatin motion (Berg et al., 2023). Lamins, the other major mechanical component of the nucleus (Dahl et al., 2004; Lammerding et al., 2006; Swift et al., 2013; Stephens et al., 2017; Hobson et al., 2020; Vahabikashi et al., 2022), are known to be added to the nucleus as it grows in size during replication and throughout interphase. Finally, actin confinement and contraction are known to be the antagonistic forces that cause nuclear deformations and rupturing (Le Berre et al., 2012; Hatch and Hetzer, 2016; Cho et al., 2019; Mistriotis et al., 2019; Nmezi et al., 2019; Xia et al., 2019; Pho et al., 2023; Alabi et al., 2025) hypothesized by the nuclear drop model (Katiyar et al., 2022; Dickinson et al., 2024). Specific changes in actin have been reported at the end of interphase due to decreased focal adhesions (Jones et al., 2018); these directly affect the actin cap (Kim et al., 2012), which confines and antagonizes the nucleus and its shape (Pho et al., 2023). These changes are more relevant in constant cycling cells, for example cancer cells, that more frequently exist in S or G2 compared to most cells, which remain in G1 or G0 state.

[1]Biology Department, University of Massachusetts Amherst, Amherst, MA 01003, USA. [2]Molecular and Cellular Biology, University of Massachusetts Amherst, Amherst, MA 01003, USA. [3]Howard Hughes Medical Institute and the Department of Biology, University of Massachusetts Amherst, Amherst, MA 01003, USA. [4]Biology Department, University of Massachusetts Dartmouth, North Dartmouth MA 02747, USA. [5]Department of Molecular and Cellular Biology, Baylor College of Medicine, Houston, TX 77030, USA. [6]Molecular, Cellular and Developmental Biology, University of Colorado Boulder, Boulder, CO 80309, USA.
*These authors contributed equally to this work

‡Author for correspondence (andrew.stephens@umass.edu)

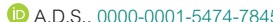 A.D.S., 0000-0001-5474-7845

Journal of Cell Science

The interphase portion of the cell cycle includes three stages. Gap 1 (G1) is the majority of the interphase cell cycle in which the nucleus exists as a diploid genome. In synthesis (S) the genome is replicated by DNA polymerase and its associated proteins. The nucleus grows during this stage independent of replication (Iida et al., 2022). S phase is followed by gap 2 (G2) in which the nucleus continues moderate growth for a short period of time and prepares to enter mitosis. Tracking cells through the interphase stages of the cell cycle can be achieved through use of fluorescent ubiquitination-based cell cycle indicator (FUCCI) cell lines that have fluorescently labeled Cdt1 and geminin, which, respectively, are expressed during G1/S and S/G2 (Sakaue-Sawano et al., 2008; Marcus et al., 2015). Furthermore, cells actively undergoing replication can be labeled with BrdU, a DNA analog (Gratzner et al., 1975). Nuclear size also correlates with interphase stage. Established cell cycle inhibitors that stall cells include lovastatin for G1 (Rao et al., 1999) and RO-3366 for G2 (Vassilev et al., 2006), and provide a means to directly modulate the interphase stages.

Using different interphase stage indicators, here, we measured how nuclear shape, integrity and mechanics are affected by the different stages of interphase. Population images revealed that nuclear blebs are present at levels similar to total population distributions. Timelapse imaging revealed that blebs formed predominantly in G1 and persisted into other interphase stages. Imaging nuclear heights revealed that G1 nuclei were under greater actin confinement (shorter nuclear height) and had more focal adhesions than G2 nuclei. When G1 and late S/G2 nuclei were placed under similar confinement using an artificial confiner, late S/G2 nuclei displayed more frequent ruptures than G1 nuclei. Finally, dual micropipette micromanipulation single nucleus force measurements confirmed that G1 nuclei were stiffer than G2 nuclei. Immunofluorescence measurements of histone modification states and lamin levels were performed to determine the basis for this nuclear stiffness change. We provide novel findings that both nuclear and actin mechanics change throughout interphase to affect nuclear blebbing and rupture, which are applicable to human disease.

## RESULTS
### Nuclear blebs are present equally across interphase cell cycle stages

A nuclear bleb is a protrusion >1 μm of the nucleus that forms during interphase and is hallmarked by loss of DNA density relative to the main nuclear body (Bunner et al., 2024; Chu et al., 2025; Pujadas Liwag et al., 2025). To determine whether nuclear blebs were overrepresented in a particular stage of the interphase portion of the cell cycle, we population imaged static wild-type (WT) and perturbed HT1080 FUCCI human cells (Marcus et al., 2015). Using FUCCI cells, we can measure interphase stage through use of fluorescent mKO–Cdt1, which increases in G1 and then is degraded gradually upon entering S phase, whereas mAG–geminin begins to be expressed in S phase and persists throughout G2 to be degraded in G1 (Fig. 1A,B). Thus, interphase stages are labeled as G1 (Cdt1 only), early S (Cdt1 and geminin) and late S/G2 (geminin only). Furthermore, we tracked FUCCI in WT cells, with low nuclear blebbing, and cells treated with chromatin decompaction drugs known to induce increased nuclear blebbing (VPA and DZNep, Fig. 1C). To determine whether nuclear blebs enrich in a specific interphase stage, we compared the percentage of cells at each stage in the total population and in cells with blebbed nuclei. In WT cells, the percentage at each stage was similar for the total population and for cells with blebbed nuclei (Fig. 1D), suggesting that nuclear blebs

do not 'prefer' a specific interphase stage. Increasing the amount of euchromatin, by treatment with the histone deacetylase inhibitor valproic acid (Gurvich et al., 2004), and decreasing the amount of heterochromatin, by treatment with the histone methyltransferase inhibitor DZNep (Miranda et al., 2009), caused increased nuclear blebbing but there were still similar cell stage distributions between total cells and cells with blebbed nuclei (Fig. 1E,F). In conclusion, blebbed nuclei are present at an equal proportion to the population across the interphase cell cycle, showing no preference in WT or perturbed FUCCI cells.

As an alternative way to measure interphase stage distributions in other cell types, we utilized BrdU incorporation to label nuclei in S phase. Again, we compared WT mouse embryonic fibroblasts (MEF) cells, with low nuclear blebbing, to cells with higher nuclear blebbing caused by chromatin (VPA and DZNep) or lamin perturbation ($Lmnb1^{-/-}$, Fig. 2A). BrdU, as assessed by immunofluorescence imaging 30 min after addition, labels nuclei actively replicating their DNA and thus in S phase (Fig. 2B). The percentage of BrdU-labeled nuclei in the total population and those with nuclear blebs were similar across all conditions. This agrees with the Fig. 1 FUCCI data, showing again that nuclear blebs are equally represented in S phase and non-S phase nuclei. Interestingly, between decreased heterochromatin perturbation (DZNep) and lamin B1 knockout ($Lmnb1^{-/-}$), the percentage of total cells in S phase changed significantly (Fig. 2C), which agrees with reports that lamin B1 loss elongates S phase (Camps et al., 2014). Even though the whole population changed, the percentage of nuclei with blebs in S phase matches the whole population. This data suggests that nuclear blebbing occurs equally across the interphase stages even when the distribution of cells undergoing DNA replication changes. HT1080 FUCCI cells labeled with BrdU showed that the percentage of cells in S phase was similar for the total cells and cells with blebbed nuclei for each condition (Fig. 2D). Together this data across two cell types and two methods reveal that nuclear blebs are presented equally throughout interphase stages.

### Nuclear blebs preferentially form in G1 and persist in S and G2

Nuclear blebs form during interphase, persist, and are the site of repetitive nuclear envelope rupture (Stephens et al., 2018, 2019b). Using time-lapse imaging to track nuclear shape, we aimed to determine when nuclear blebs form during the interphase stages of the cell cycle. HT1080 FUCCI cells were imaged for 36 h to ensure tracking of cells from start to finish of interphase (Fig. 3A). Time-lapse imaging revealed that nuclear blebs formed preferentially in G1 because at any given time 50% of cells are in G1 but 80% of blebs form in G1, a statistically significant increase and clear overrepresentation above population behavior for both WT and upon VPA-based chromatin decompaction (Fig. 3A–C; Movie 1). Most nuclei that formed a nuclear bleb ruptured soon after (85% rupture, 17/20 blebs WT), which agrees with previous literature (Stephens et al., 2019b). These blebs persisted into early S and late S/G2, as reported by FUCCI fluorescence (Fig. 3A), indicating that the equal distribution of nuclear blebs across the cell cycle stems from the formation of a bleb in G1 and its persistence throughout S and G2 phases.

We time-lapse imaged other cell lines to determine whether over representation of nuclear bleb formation in G1 could be recapitulated. Imaging MEF NLS–GFP nuclei normalized to time in the cell cycle recapitulated the formation of nuclear blebs occurring predominantly in the first 50–60% (mostly G1) of the

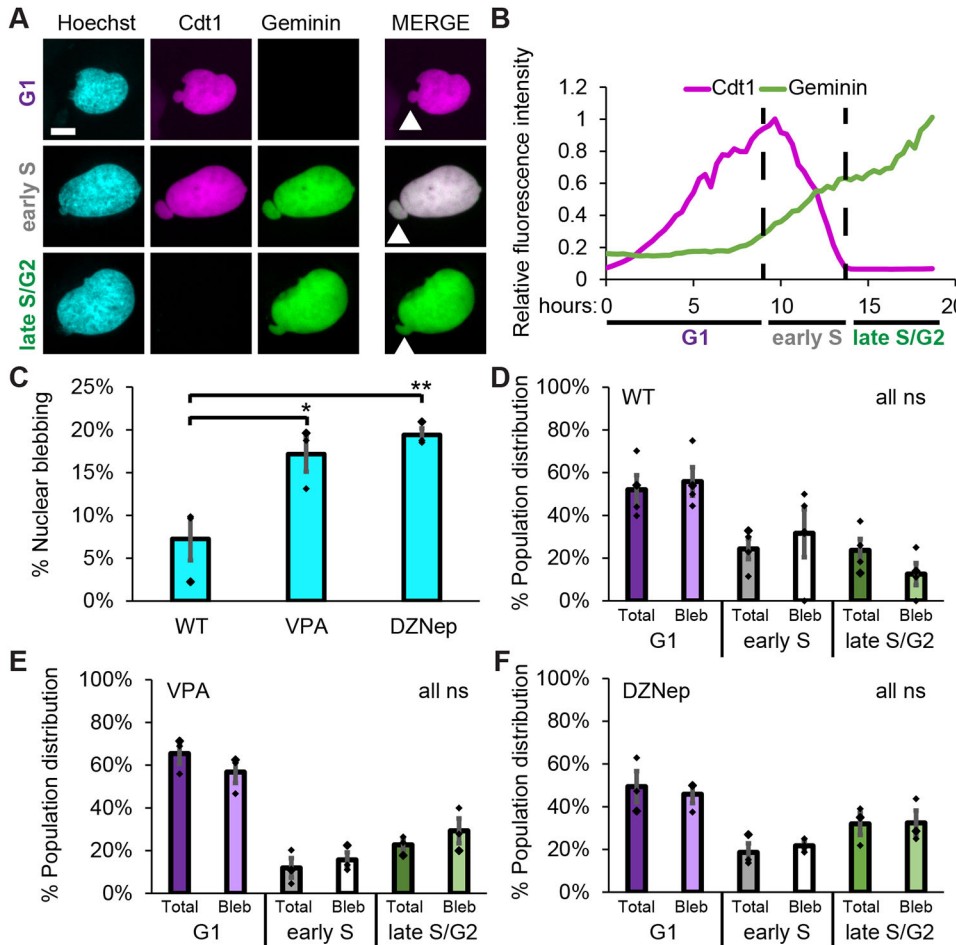

**Fig. 1. Static imaging of FUCCI cells reveals that nuclear blebs are equally distributed throughout interphase stages.** (A) Example images of HT1080 FUCCI cells with Cdt1 and geminin showing G1 (purple), early S (gray), and late S/G2 (green). White arrowhead denotes a nuclear bleb in the merged image. (B) Example graph of relative Cdt1 and geminin intensity throughout the interphase part of the cell cycle for a single cell measured in hours. A relative intensity of 1 represents maximum intensity. (C) Graph of percentage of cells displaying a nuclear bleb in WT control cells and WT cells after chromatin decompaction through treatment with HDACi VPA or HMTi DZNep. Biological triplicates of $n>100$ cells each. (D–F) Graphs of the percentage of cells in each interphase stage for the total population of cells (darker color) and for only cells presenting a nuclear bleb (lighter color) for WT control (D), and WT cells after chromatin perturbations that increased nuclear blebbing by increasing euchromatin through treatment with VPA (E) or decreased heterochromatin through treatment with DZNep (F). Results are for three or four biological replicates with a total of >400 cells for the population and >20 cells with nuclear blebs. *$P<0.05$; **$P<0.01$; ns, not significant, $P>0.05$ (unpaired two-tailed Student's $t$-test). Error bars represent s.e.m. Scale bar: 10 µm.

interphase cell cycle and then these persisting throughout interphase (Fig. S1A). Time-lapse imaging of RPE1 FUCCI cells showed a similar trend to HT1080 cells, with a higher frequency of cells with nuclear blebs being in G1 than the percentage of general cell population at G1 stage (Fig. S1B). Thus, dynamic imaging of nuclear bleb formation in multiple cell types consistently showed an increased frequency of forming a nuclear bleb in G1 and that the blebs persist into other interphase cell cycle stages.

## G1 nuclei are under greater actin confinement than late S/G2 due to focal adhesions

Actin confinement and actin contraction are two major factors that antagonize nuclear shape and cause nuclear blebbing (Le Berre et al., 2012; Hatch and Hetzer, 2016; Stephens et al., 2018; Cho et al., 2019; Mistriotis et al., 2019; Nmezi et al., 2019; Xia et al., 2019; Pho et al., 2023). Thus, we assayed actin confinement by nuclear height and actin contraction by active actomyosin by assessing the levels of phosphorylated myosin light chain 2 (pMLC2; MLC2 is also known as MYL2). Confocal imaging of live HT 1080 FUCCI cells revealed that nuclear height in G1 nuclei 5.2±0.2 µm (mean±s.e.m.) was significantly decreased (actin confinement increased) relative to taller late S/G2 nuclei, at 6.0±0.2 µm, for WT (Fig. 3D,E). To directly test whether actin confinement determines nuclear height, cells were treated with the actin depolymerization drug latrunculin A for 1 h, which drastically increased nuclear height and abolished the differences between G1 and late S/G2 nuclei (Fig. 3E, +Lat A). Actin depolymerization also resulted in no new nuclear blebs forming over 2 h (Fig. S1D), in

agreement with previous publications cited above. HT1080 FUCCI VPA-treated cells show a decreased G1 nuclear height relative to WT of from 5.2 to 4.0±0.1 µm, which agrees with previous reports of decreased nuclear height (increased actin confinement) and is consistent with reported decreased chromatin rigidity (Berg et al., 2023; Pho et al., 2023). These more confined VPA-treated cells recapitulated the increased nuclear height from G1 nuclei at 4.0±0.1 µm to late S/G2 nuclei at 4.8±0.2 µm ($P<0.001$, Fig. 3F), denoting decreased actin confinement. RPE1 FUCCI cells recapitulated shorter G1 nuclei (more actin confinement) and taller late S/G2 nuclei (less actin confinement, Fig. S1C). These data agree with reports of increased actin confinement in G1 (Aureille et al., 2019). Measurements of actin contraction by assessing pMLC2 revealed no significant change in active actomyosin between the different stages of interphase (Fig. S2A–C). Thus, G1 nuclei are under increased actin confinement compared to late S/G2 nuclei, whereas the actin contraction remains similar across the interphase stages.

Actin confinement relies on the cell attaching by focal adhesions to the substrate (Kim et al., 2012), and these focal adhesions are removed in G2 in preparation for cell rounding in mitosis (Jones et al., 2018). To quantify the number of focal adhesions in each interphase stage, we fixed FUCCI cells and performed immunofluorescence for the focal adhesion protein paxillin (Fig. 3G). G1 nuclei measured a statistically significant increase in both focal adhesion density and area relative to cell size compared to late S/G2 nuclei (Fig. 3H,I). Thus, G1 cells have more focal adhesions, providing a plausible mechanism for increased actin confinement leading to increased nuclear bleb formation.

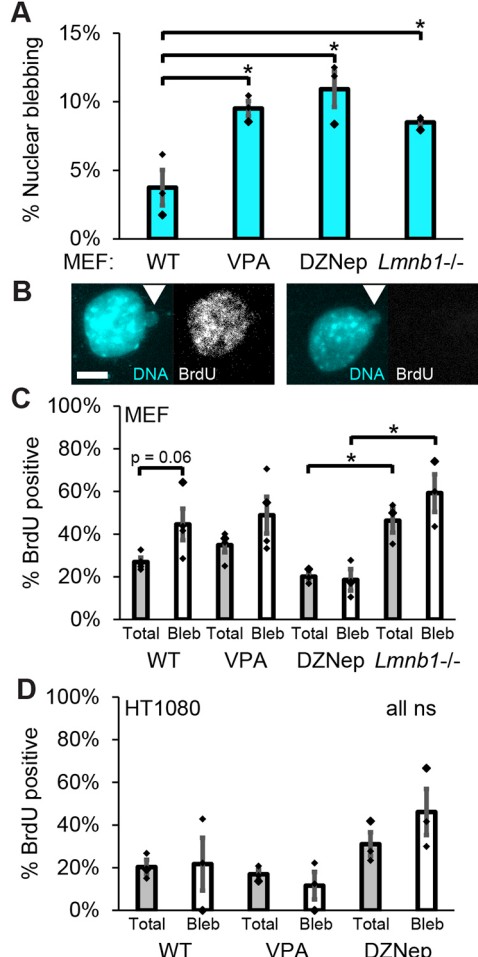

**Fig. 2. Nuclear blebs are present at population levels in replicating cells.** (A) Graph of the percentage of cells displaying nuclear blebbing in MEF WT cells and WT cells after chromatin perturbations through treatment with VPA or DZNep, and in cells with loss of lamin B1 (*Lmnb1*$^{-/-}$). Results are for biological triplicates of $n>140$ cells each. (B) Example images of a blebbed nucleus positive (left) and negative (right) for DNA analog BrdU that denotes cells undergoing replication. White arrowhead denotes nuclear bleb. (C) Graph of the percentage of MEF total cells (gray) and blebbed nuclei (white) that are positive for BrdU staining, and thus in S phase. (D) Graph of the percentage of HT1080 total cells (gray) and blebbed nuclei (white) that are positive for BrdU staining, and thus in S phase. Results are for three or four biological replicates with each replicate containing $n>160$ cells each and $n>7$ blebs for WT and $n>17$ blebs for every other condition. *$P<0.05$; ns, not significant, $P>0.05$ (unpaired two-tailed Student's *t*-test). Error bars represent s.e.m. Scale bar: 10 μm.

Focal adhesions can be modulated through use of established focal adhesion kinase inhibitor (FAKi) such as PF-573228 (Slack-Davis et al., 2007; Wang et al., 2016). We used this FAKi to determine whether higher amounts of focal adhesions in G1 drive increased nuclear confinement and blebbing. HT1080 FUCCI cells treated with FAKi results in a significant decrease in the number of focal adhesions per cell area, as measured by paxillin (Fig. 4A,B). Nuclear height differences between G1 and late S/G2 nuclei were eliminated, as G1 nuclei treated with FAKi were significantly taller than untreated G1 nuclei (Fig. 4C). Finally, treatment with FAKi significantly decreased nuclear blebbing in HT1080 nuclei treated with VPA (Fig. 4D). Similarly, FAKi treatment of MEF cells resulted in decreased focal adhesions, nuclear height and nuclear blebbing but no change in pMLC2 (Fig. S3). Thus, recapitulating

focal adhesion loss from G1 to late S/G2 through FAKi treatment in G1 nuclei resulted in similar decreased nuclear confinement (increased nuclear height) and decreased nuclear blebbing.

## Artificial compression shows that nuclei in G2 are more fragile than in G1

One hypothesis is that G1 nuclei, which natively are more likely to form a bleb, will show more nuclear ruptures under increased artificial confinement. An alternative hypothesis is that G2 nuclei will rupture more because they are weaker but usually under less confinement due to alleviation of focal adhesions. To test these hypotheses directly, we artificially modulated confinement by using a compression device comprising a coverslip with micropillars of defined heights (Le Berre et al., 2012). This experiment provides the ability to compress nuclei of cells in different interphase stages to the same height.

FUCCI cells and artificial confinement provide the ability to visualize how neighboring asynchronous cells from different interphase stages respond to specific confinement heights. Measurements of WT cells revealed that G1 nuclei are 5.2±0.2 μm (mean±s.e.m.) in height (Fig. 3E). Thus, we placed the nuclei under increased artificial confinement to 4 μm (Fig. 5A). Under artificial confinement of 4 μm G1 nuclei displayed nuclear rupture in 17±5% (mean±s.e.m.) of cells (Fig. 5B; Movie 2). However, late S/G2 nuclei displayed a significant increase in nuclear rupture under the same 4 μm artificial confinement, to 49±8%. This difference in rupture percentages was not due to changes in percentage size increase in two-dimensional area upon confinement ($P>0.05$, Fig. 5C). To show that confinement does determine nuclear rupture, we then placed nuclei under increased artificial confinement to 3 μm, which resulted in increased nuclear rupture relative to 4 μm confinement for both G1 and late S/G2 nuclei (Fig. 5B). Upon this drastic increase in artificial confinement, G1 and late S/G2 nuclei showed a statistically similar 85–96% nuclear rupture. Furthermore, upon applying 3 μm artificial confinement, late S/G2 nuclei ruptured before neighboring G1 nuclei (Fig. 5D). Thus, late S/G2 nuclei are more susceptible to rupture than G1 nuclei under similar confinement.

## Micromanipulation force measurements confirm G2 nuclei are weaker than G1

Nuclear blebbing and rupture stems from loss of nuclear stiffness from chromatin decompaction and/or loss of lamins (Kalukula et al., 2022). To aid nucleus isolation from the cytoskeleton for dual micropipette micromanipulation force measurements, we used MEF vimentin-null nuclei (Stephens et al., 2017; Currey et al., 2022). By isolating a single nucleus and pulling on it in a micromanipulation experiment while measuring forces, we can separate the nuclear mechanical contributions of chromatin and lamins. During micromanipulation, the pull pipette extends the nucleus while deflection of the force pipette multiplied by its pre-measured bending constant provides a measure of force. Chromatin rigidity determines the short extension regime (0–3 μm extension), whereas lamins determine strain stiffening that occurs at higher extensions >3 μm (Banigan et al., 2017; Stephens et al., 2017; Currey et al., 2022). This force versus extension relation can be graphed, with the slope providing a nuclear spring constant nN/μm (Fig. 6A).

To determine which nuclei were in G1 and G2, we measured nuclear size, which relatively doubles throughout interphase. We compared nuclear size distributions upon use of drug inhibitors to stall the cell cycle (Iida et al., 2022), which shifted to smaller nuclei upon stalling cells in G1 through treatment with lovastatin and

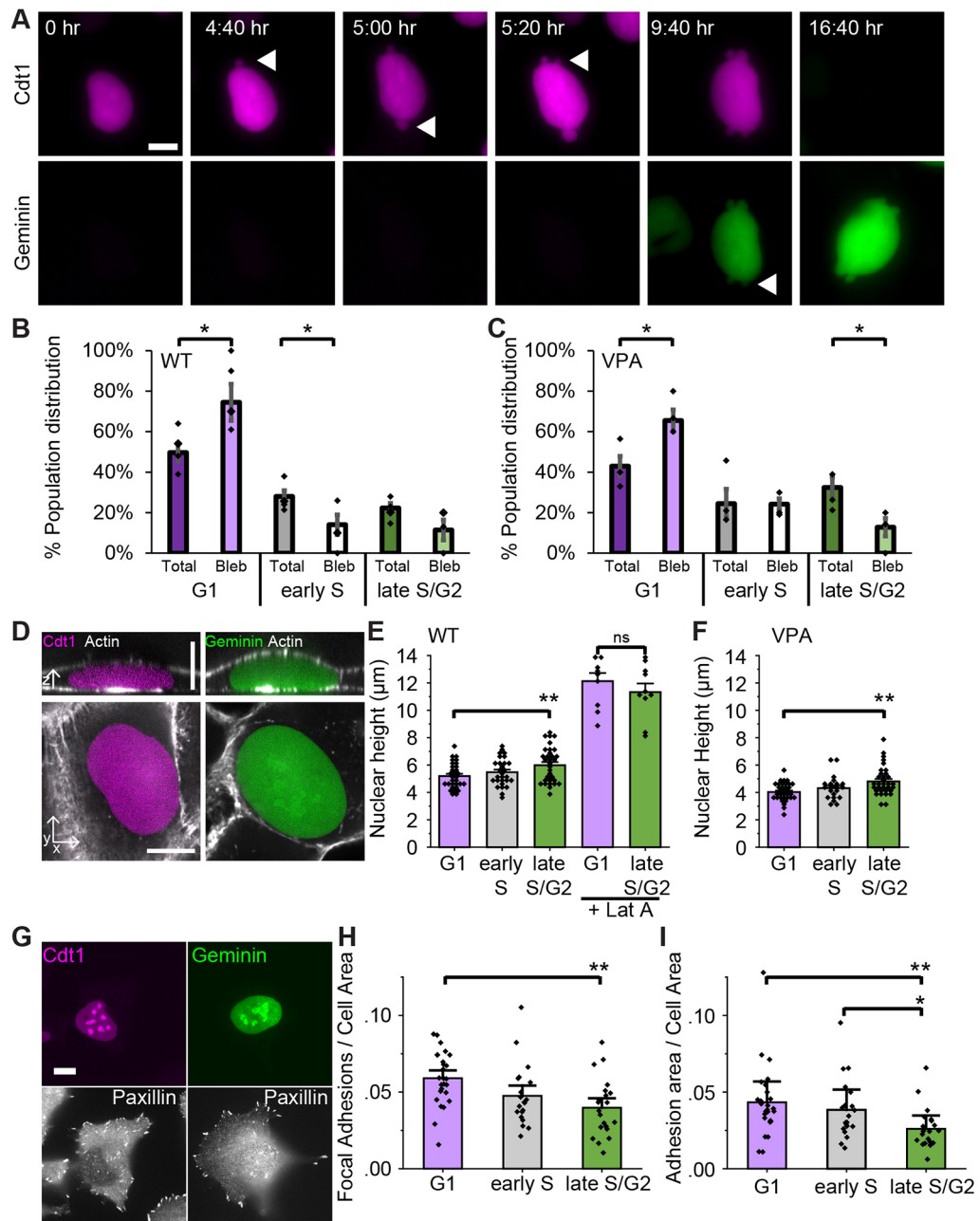

**Fig. 3. Time-lapse imaging reveals that nuclear blebs form predominantly in G1 due to actin confinement and then persist throughout interphase.** (A) Example images of a HT1080 FUCCI nucleus forming blebs, denoted by arrowheads, in G1 and those blebs persisting into late S/G2. Time in hours:mins (hr) throughout interphase denoted in white in the upper left of each image. (B,C) Graphs of the percentage population of total cells (darker) and cells in which a nuclear bleb formed (lighter) in G1, early S, or late S/G2 for WT control (B), and WT cells after chromatin decompaction through VPA treatment (C). Results are for three or four biological replicates, respectively, where each replicate contains $n>180$ cells and $n>8$ nuclear blebs. MEF and RPE1 cells showed similar results for increased bleb formation relative to the population (see Fig. S1). (D) Example images of nuclear height in G1 (purple) and late S/G2 (green) to measure actin confinement. (E,F) Graph of nuclear height for G1, early S and late S/G2 in WT control and WT cells after treatment with the actin depolymerization drug latrunculin A (+Lat A) for 1 h (E), or after treatment with VPA (F). Graphs are an average of $n=31$, 28, 32, respectively, for WT G1, early S and late S/G2; without LatA and with LatA G1 and late S/G2 are both $n=10$; $n=34$, 21, 33, respectively, for VPA-treated nuclei. RPE1 nuclei showed similar actin height changes from G1 to late S/G2 (see Fig. S1). (G) Example images of G1 and late S/G2 nuclei with immunofluorescence of paxillin focal adhesions. (H,I) Graphs of (H) paxillin focal adhesions per cell area and (I) adhesion area per cell area in WT nuclei in G1, early S and late S/G2. $n=24$, 20, 21, respectively. Actin contraction was similar throughout interphase (see Fig. S2). *$P<0.05$; **$P<0.01$; ns, not significant, $P>0.05$ (unpaired two-tailed Student's $t$-test). Error bars represent s.e.m. Scale bars: 10 µm.

larger nuclei upon stalling cells in G2 stall through treatment with RO-3306 compared to WT control (Fig. 6B). Using this data in asynchronous cell culture, we selected smaller nuclei, of 100–200 µm² size, to denote G1, and larger nuclei of >300 µm²

size, to denote G2. Relative to smaller nuclei, larger nuclei were significantly weaker, showing a decrease in the nuclear spring constant from 0.54±0.03 nN/µm (mean±s.e.m.) for the chromatin-dominated short regime in small nuclei compared to

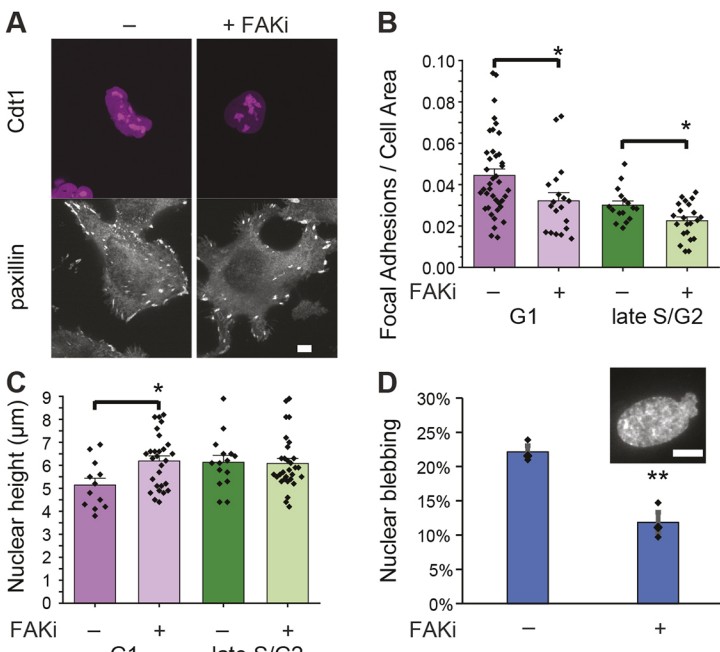

**Fig. 4. Inhibition of focal adhesion kinase decreases focal adhesions, nuclear confinement and nuclear blebbing.**
(A) Example images of HT1080 FUCCI G1 nuclei, as determined by Cdt1 fluorescence and immunofluorescence of paxillin focal adhesions. (B) Graph of paxillin focal adhesions per cell area in G1 and late S/G2 nuclei without (−) or with (+) 5 µM FAKi PF-573228. n=40, 19, 16, 22, respectively. (C) Graph of nuclear height for G1 and late S/G2 cells treated without or with 5 µM FAKi PF-573228 (n=12, 28, 15, 28, respectively). (D) Graph of nuclear blebbing percentages in HT1080 FUCCI treated with VPA and without (−) or with (+) 5 µM FAKi PF-573228. Results are for one biological replicate with n>200. *P<0.05; **P<0.01 (unpaired two-tailed Student's t-test). Error bars represent s.e.m. Scale bars: 10 µm.

0.35±0.02 nN/µm in large nuclei (Fig. 6C,D). Interestingly, long extension nuclear spring constant, which is composed of both chromatin and lamin contributions also significantly decreased in larger nuclei (1.10±0.04 to 0.60±0.03 nN/µm, Fig. 6E). This decrease could be due to the large contribution of chromatin-based rigidity. To measure lamin-based strain stiffening, we subtracted the nuclear spring constant for the short extension regime from that for the long extension regime. Lamin-based strain stiffening also decreased from small to large nuclei (0.56±0.03 to 0.24±0.03 nN/µm, Fig. 6F). Thus, both chromatin- and lamin-based contributions to nuclear rigidity decreased from stiffer and smaller nuclei representing G1 to weaker and larger nuclei representing G2. This confirms the results from the artificial confinement experiment where G2 nuclei ruptured at a greater frequency than G1 under the same confinement.

Next, we used cell cycle stalling drugs to synchronize cells in G1 with lovastatin and G2 with RO-3366. We performed a similar set of micromanipulation force measurements, which returned near identical values for the asynchronous small and larger nuclei used as proxies for G1 and G2. Nuclei from lovastatin-treated cells, used to cause a G1 stall, showed a statistically similar stiffer value for the short, long and strain stiffening regimes that was comparable to that for small nuclei. Nuclei from RO-3366-treated cells, to stall in G2, gave a statistically significant weaker value in the short, long, and strain stiffening regimes than for lovastatin G1 stalled nuclei, similar to large nuclei. Thus, relative to lovastatin G1 stalled nuclei, RO-3366 G2 stalled nuclei had significantly weaker values for the chromatin-based short regime, chromatin- and lamin-based long regime, and the lamin-based strain stiffening nuclear spring constants (Fig. 6D–F). Taken together, we provide novel evidence that G2 nuclei are weaker than G1 nuclei due to weakening of both major nuclear stiffness components, chromatin and lamins.

To determine the source of changes in both the chromatin and lamin nuclear spring constants, we measured chromatin histone modifications and lamin levels through immunofluorescence. In MEF cells, the levels of the constitutive heterochromatin marker H3K9me3 decreased from G1 to G2 nuclei (Fig. 6H) whereas lamin A/C levels remained similar (Fig. 6H). We recapitulated these

findings in HT1080 FUCCI human cells where constitutive heterochromatin decreases while lamin A/C remains the same from G1 to late S/G2 (Fig. S4). Thus, the decrease from G1 to G2 in chromatin-based nuclear stiffness could be due to decreased H3K9me3 levels, whereas the reason for the decreased lamin-based nuclear stiffness remains unclear.

## Loss of H3K9me3 can account for decreased chromatin and lamin nuclear stiffness from G1 to G2

Chromatin is known to connect with lamins in cells (Harr et al., 2015; van Schaik et al., 2020). Biophysical modeling suggests these chromatin–lamin connections are necessary mechanical linkers for the two-regime behavior of the nucleus (Banigan et al., 2017; Strom et al., 2021). The constitutive heterochromatin marker H3K9me3 binds to lamins, has peripheral localization and also shows internal localization to chromocenters which are dense chromatin foci (Manning et al., 2025; Fig. 7A). Using spinning disk confocal imaging, we find that relative to smaller nuclei (G1), larger nuclei (G2) have both decreased peripheral and whole-nucleus H3K9me3 levels (Fig. 7A–C). The loss of peripheral H3K9me3 from G1 to G2 occurs in all cell types and conditions while also being independent of focal adhesions (Fig. S5). The loss of H3K9me3 specifically from G1 to G2 could result in loss of chromatin–lamin linkers previously modeled to affect both the chromatin regime and the lamin strain stiffening response (Strom et al., 2021). Analysis of chromocenters shows that the intensity and size of chromocenters is similar in small and large nuclei (Fig. 7D). Thus, loss of peripheral and whole-nucleus H3K9me3 could account for decreased nuclear stiffness from both the chromatin and lamin regime from G1 to G2.

H3K9me3 levels can be decreased by treatment with the SUV39H1 methyltransferase inhibitor chaetocin (Greiner et al., 2005). We have recently shown that this loss of H3K9me3 decreases both chromatin and lamin-based nuclear stiffness in smaller G1-like nuclei, resulting in increased nuclear blebbing (Manning et al., 2025). To show this effect, we reanalyzed the data in the previous study as relative change in nuclear spring constant. This analysis indicates that loss of H3K9me3 through chaetocin treatment

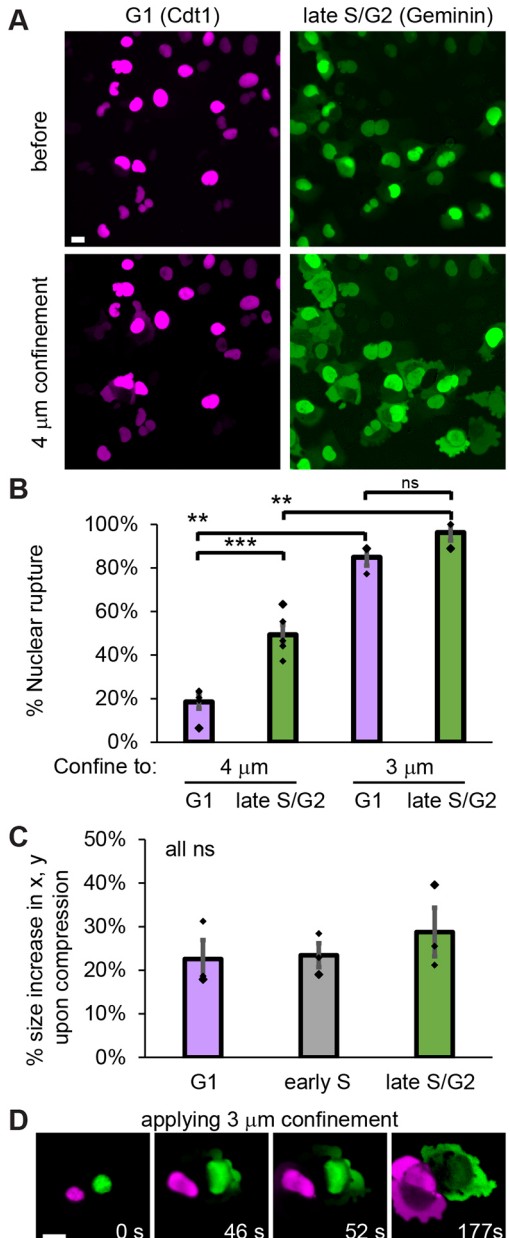

**Fig. 5. Late S/G2 nuclei rupture more frequently than G1 nuclei when placed under artificial confinement.** (A) Example images of a WT HT 1080 FUCCI labeled nuclei, as determined by Cdt1 (only=G1) and geminin (only=late S/G2) before and after 4 µm artificial confinement. (B) Graph of the percentage of G1 and late S/G2 nuclei that rupture upon artificial confinement to 4 µm and 3 µm. (C) Graph of the percentage increase in nuclear size measured in x,y upon 4 µm artificial confinement in WT HT 1080 cells in G1, early S and late S/G2. Results in B and C are for five replicates each with *n*>50 cells for 4 µm and *n*>30 cells for 3 µm artificial confinement. (D) Example image of cells progressively undergoing artificial confinement to 3 µm over time showing that the late S/G2 nucleus ruptures before the neighboring G1 nucleus. Time denoted in white as seconds (s). *$P<0.05$; **$P<0.01$; ***$P<0.001$; ns, not significant, $P>0.05$ (unpaired two-tailed Student's *t*-test). Error bars represent s.e.m. Scale bars: 10 µm.

decreases the short extension, long extension and strain stiffening (long–short) regime values (Fig. 7E), which recapitulated nuclear spring constant changes from G1 to G2 (Fig. 6). These data are consistent with mechanical modeling of loss of chromatin to lamin linkages, which predicts deceases in both the chromatin short

extension regimes and the lamin long extension strain stiffening (Strom et al., 2021). Taken together the data show that the measured loss of H3K9me3 from G1 to G2 nuclei is sufficient to alter both the chromatin- and lamin-based nuclear stiffness regimes.

## DISCUSSION

### G1 nuclei are both stiffer and under greater actin antagonism

Our data reveals that nuclear blebs form in G1 at an over-represented rate relative to the total population of cells, and this leads to nuclear ruptures which cause cellular dysfunction. Moreover, a force balance between actin antagonism and nuclear resistance is necessary to maintain nuclear homeostasis. Thus, changes in either nuclear mechanics and/or actin or external confinement can result in increased nuclear bleb formation in G1/G0 nuclei. In theory, the ability for G1 nuclei to form a nuclear bleb might start the cell and nucleus down a path of dysfunction that contributes to a worsening disease state.

Our data provides a unique insight into when the nucleus is rigid and actin antagonism is high. Increased levels of focal adhesions aid increased actin-based nuclear confinement, providing sufficient antagonism to overcome a stiffer G1 nucleus, as evidenced by the disproportionate increase in nuclear bleb formation in G1 nuclei and ruptures (Figs 3, 4 and 6). Removal of focal adhesions alleviates nuclear confinement and blebbing (Fig. 4; Fig. S3). From our previous work, the strength of actin confinement can be seen in nuclei with decompacted chromatin mediated by VPA, in which the actin compresses the nuclear height significantly by 0.5 µm, or ~10% of overall nuclear height, due to the weakening of the chromatin-based nuclear stiffness (Berg et al., 2023; Pho et al., 2023). The importance of nuclear confinement is further confirmed by artificial compression (Fig. 5). A change from 4 µm artificial confinement to 3 µm confinement resulted in a significant increase in nuclear ruptures to nearly 100% (84–96%) of all nuclei deforming and rupturing independent of nuclear interphase stage and strength. Thus, our data suggests that even a rigid nucleus will deform and rupture under high levels of actin confinement.

### S phase DNA replication does not affect nuclear blebbing but might impact dysfunction from nuclear blebbing

The action of DNA replication has no significant impact on nuclear blebbing. Our data clearly show that in S phase nuclear blebs are present at population levels while new blebs form below population distribution levels, which means there is no support for DNA replication having a role in nuclear bleb formation (Figs 1–3). Although DNA replication has no measurable impact on nuclear blebbing, it has been reported to impact dysfunction associated with nuclear blebbing and rupture after DNA damage. The literature remains contentious on the importance of nuclear rupture causing increased DNA damage. Many studies have shown that deformation of the nucleus without rupture is sufficient to induce increased DNA damage (Denais et al., 2016; Raab et al., 2016; Xia et al., 2018). However, increased DNA damage caused by deformation only or by rupture are reported to be reliant on DNA replication (Irianto et al., 2017; Shah et al., 2021; Chu et al., 2025).

Although DNA replication and its associated S phase have no apparent role in nuclear bleb formation, the result of going through this process leads to G2, and its effects on both nuclear and actin mechanics. One hypothesis is that cancer cells that are constantly going through the cell cycle might be more susceptible to nuclear blebbing and rupture due to passing through the interphase stages of DNA replication (S phase) and onto G2. Thus, our data shows that S phase does not significantly alter nuclear blebbing.

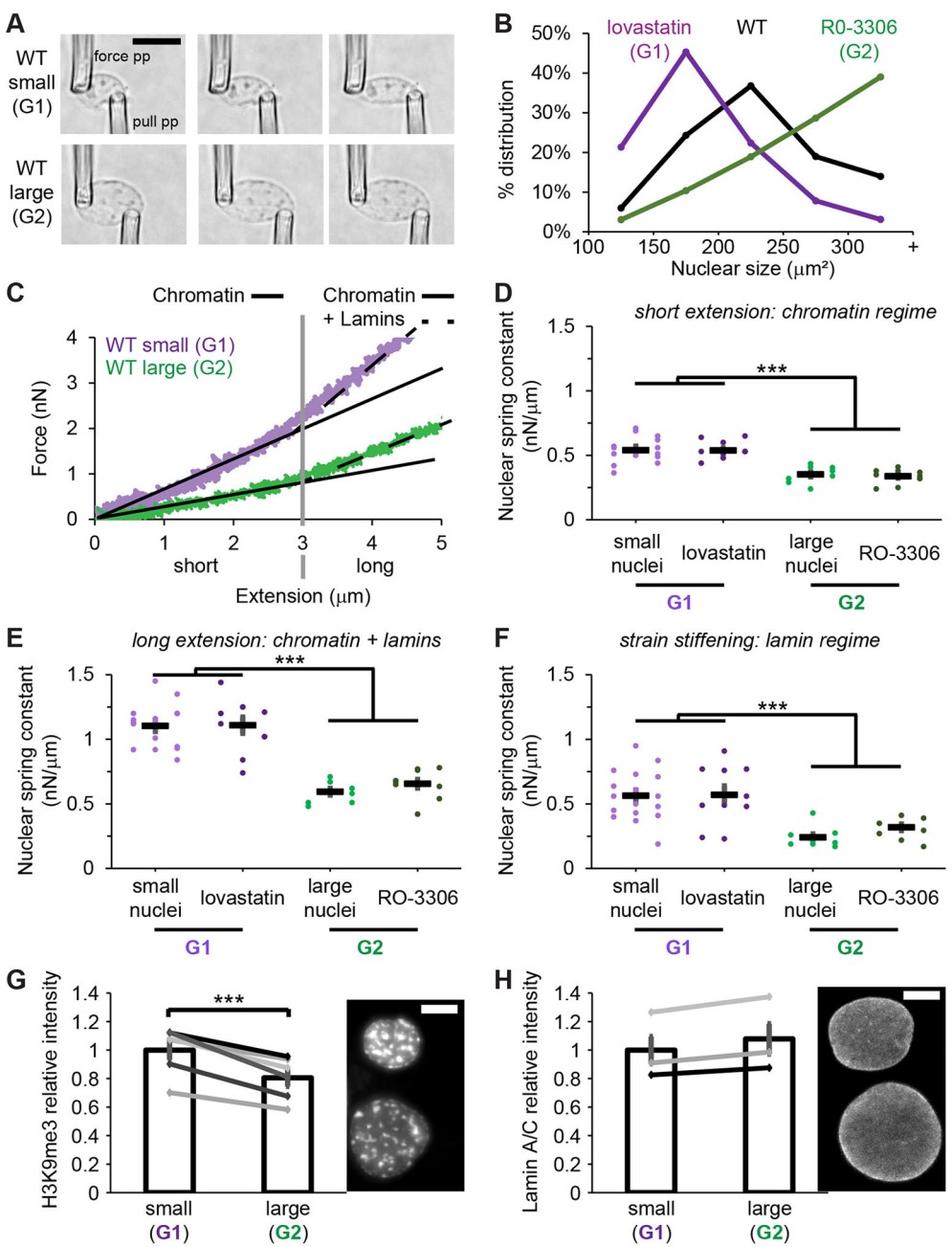

**Fig. 6. Micromanipulation nuclear force measurements confirm that G2 nuclei are mechanically weaker than G1 nuclei.** (A) Example images of a small G1 and large G2 nuclei during micromanipulation force extension measurements. The pull pipette extends the nucleus as deflection of the force pipette multiplied by its pre-measured bending constant provides a measure of force. (B) Graph of percentage distribution of total population by nuclear size after lovastatin treatment to stall cells in G1 ( purple, *n*=188), WT cells (black, *n*=632), and after R0-3306 treatment to stall cells in G2 (green, *n*=186). (C) Example graphs of force in nanonewtons versus extension in micrometers measurement shown for small and large MEF vimentin-null nuclei representing G1 and G2, respectively. The slope of the line provides a nuclear spring constant (nN/µm) averaged for each nucleus over three force versus extension pulls. The short extension (<3 µm) measures the mechanical contribution of chromatin (solid line), whereas the long extension (>3 µm) shows strain stiffening due to lamins (dashed line). (D–F) Graphs of nuclear spring constants for (D) a chromatin-based short extension, (E) a long extension, which accounts for both chromatin and lamin contribution, and (F) lamin-based strain stiffening at longer extensions. Nuclear spring constants are reported for WT small nuclei, cells that are stalled in G1 through lovastatin treatment, WT large nuclei, and cells that are stalled in G2 stall through R0-3306 treatment (*n*=17, 11, 8, 11). (G,H) Example images and graphs relative immunofluorescence intensity of paired small and large nuclei for (G) the constitutive heterochromatin marker H3K9me3 and (H) lamin A/C. *n*=6 H3K9me3 and *n*=3 lamin A/C biological replicates, where each contained >20 nuclei for each condition. The average intensity in small (G1) nuclei was scaled to 1 to provide a relative intensity. ***P<0.001 [unpaired (D–F) or paired (G,H) two-tailed Student's *t*-test]. Error bars represent s.e.m. Scale bars: 10 µm.

## G2 nuclei are mechanically weaker but under less actin confinement

We find overall that G2 nuclei are weaker and thus lack the ability to maintain nuclear shape and integrity in the face of actin and/or external confinement. This finding has strong implications for human diseases where cells might be cycling and thus be in S and G2 more often than non-cycling cells in a G1/G0 state. Our findings reveal that G2 nuclei are more fragile under artificial

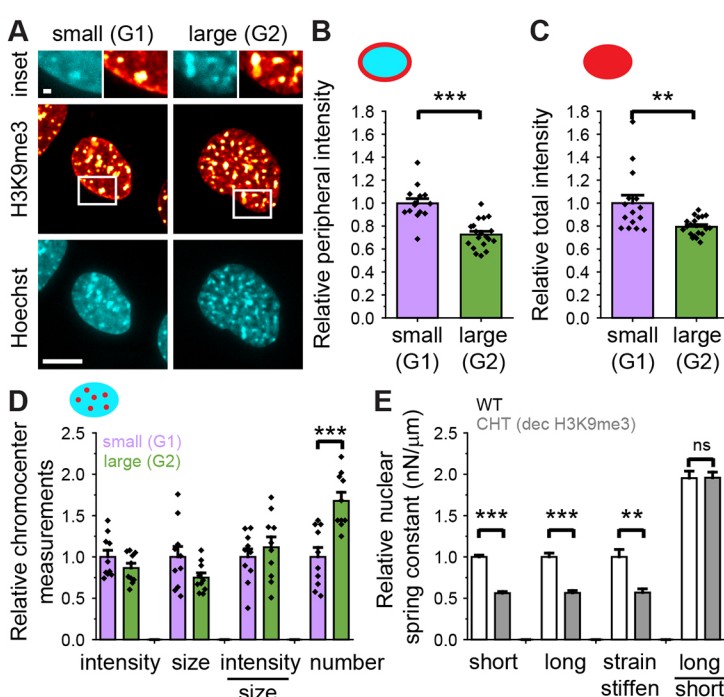

**Fig. 7. Loss of H3K9me3 occurs in the interphase cell cycle and causes a decrease in both chromatin- and lamin-based nuclear spring constant.** (A) Example confocal images of a nucleus stained for DNA with Hoechst 33342 (cyan) and constitutive heterochromatin H3K9me3 (red). The magnified area (top) highlights the differences in peripheral H3K9me3. Graphs of relative (B) peripheral and (C) whole nucleus H3K9me3 intensity for smaller (100–200 μm²) and larger (>300 μm²) nuclei representing, respectively, G1 and G2 nuclei (n=15 and 19). Similar decreases in whole nucleus and periphery H3K9me3 are also seen in HT1080 and MEF vimentin-null cells (Figs S4 and S5). (D) Graph of relative chromocenter H3K9me3 intensity, size, intensity/size, and number for smaller (100–200 μm²) and larger (>300 μm²) nuclei representing, respectively, G1 and G2 nuclei (n=10 and 10). (E) Graph of relative nuclear spring constant for a single nucleus micromanipulation force measurement for WT cells and cells treated with the SUV39H1 inhibitor chaetocin to decrease H3K9me3. This graph is from previously published data (Manning et al., 2025). Relative comparisons for each the short nuclear stiffness regime (chromatin-based), long nuclear stiffness regime (chromatin+lamin A regime) and strain stiffening regime (lamin-based, long minus short regime), and the ratio of results for the long versus short regime (n=13 and 10, respectively). The average intensity or spring constant in small (G1) nuclei was scaled to 1 to provide a relative intensity. **P<0.01; ***P<0.001; ns, not significant, P>0.05 (unpaired two-tailed Student's t-test). Error bars represent s.e.m. Scale bars: 10 μm.

confinement (Fig. 5), which is supported by the decreased nuclear stiffness seen after dual pipette micromanipulation (Fig. 6). In agreement, other publications have also reported similar findings. In glioblastoma human cell lines, U251MG cells bleb and rupture more than U87MG cells. U251MG has no p21 and is thus cycling more than G1 U87MG (Kamikawa et al., 2023). That paper concludes that G1 cells are more resistant to nuclear envelope stress than S/G2.

Nuclear stiffness decreases for both the chromatin and lamin regimes. Our direct data shows that this change in chromatin stiffness is due to loss of the constitutive heterochromatin marker H3K9me3. However, the data is less clear about why lamin-based strain stiffening is decreased because there is no significant loss of lamin A/C. Chromatin–lamin interactions are the other sub-component of nuclear force response. Physics modeling of nuclear mechanics shows that loss of chromatin–lamin interactions would result in changing both chromatin and lamin-based nuclear stiffness (Strom et al., 2021). H3K9me3 interacts with the lamina through PRR14 tethering and HP1a (Kiseleva et al., 2023). H3K9me3 nuclear periphery enrichment is significantly lost from G1 to G2, suggesting a mechanism for strain stiffening-based mechanical changes. Upon depletion of H3K9me3 through use of the SUV39H1 inhibitor chaetocin, which mimics loss of whole nucleus and peripheral H3K9me3, nuclear stiffness is lost for smaller G1-like nuclei in both the chromatin short regime and lamin-based long extension strain stiffening (Fig. 7; Manning et al., 2025). This data importantly shows that the change in nuclear stiffness is not due to nuclear size changes from G1 to G2 but instead due to loss of H3K9me3. Thus, loss of heterochromatin, which acts as a chromatin–lamin linker, is sufficient to recapitulate changes in nuclear stiffness from G1 to G2.

Although we provide a plausible mechanism for cell cycle-based decreased nuclear stiffness, other factors might also contribute to the loss of chromatin–lamin linkers. DamID data reveals that lamin-associated domains have cell cycle-specific behaviors (van Schaik et al., 2020). Telomere proximal chromatin is associated with the lamina in G1 but detaches over the cell cycle whereas centromere-

proximal chromatin accumulates over the cell cycle. It has been hypothesized that changes in telomere or end chromosome association detachment are due to changes in LAP2α (Dechat et al., 2004). Overall, similar to loss of focal adhesions in lieu of cell rounding in mitosis, loss of chromatin to lamin connections makes sense to both condense the chromatin into chromosomes and prepare for nuclear envelope breakdown at the onset of mitosis.

Chromatin–chromatin linkages are a nuclear stiffness sub-component that have been shown to be essential to nuclear mechanical response in simulations (Banigan et al., 2017; Strom et al., 2021). Our laboratory and others have shown that the local chromatin linker HP1α is a key mechanical component determining nuclear shape (Strom et al., 2021; Williams et al., 2024). HP1α is reported to be phosphorylated during the cell cycle in order to localize to the kinetochore (Chakraborty et al., 2014). This might decrease HP1α across the genome, leading to weakening of the chromatin gel polymer interior. We have also shown that long distance chromatin interactions, as assessed by HiC, provide chromatin mechanical stiffness to the nucleus (Belaghzal et al., 2021). Regarding this point, the cell cycle dynamics of chromosomal organization at single-cell resolution shows that chromatin HiC contacts in G1 are mostly long-range interactions at 50 Mb, which dramatically shifts in G2 to shorter range interactions, at 250 kb (Nagano et al., 2017). Thus, it is possible that moving from long range to shorter range interactions might also contribute to decreased chromatin-based nuclear mechanics. Overall, it remains unclear how or why a duplicated genome would scale chromatin–chromatin interactions for a transient period before mitosis.

Furthermore, actin focal adhesions are removed in lieu of cell rounding in mitosis. Decreased actin confinement in G2 nuclei results in less (or at population) levels of nuclear bleb formation even though the nucleus is weaker in G2 (Fig. 3A–F). Decreased actin confinement from G1 to G2 has been confirmed by others in HeLa, MEF and MRC-6 cells (Aureille et al., 2019). This is further supported by decreased traction forces during cell cycle progression (Vianay et al., 2018). The mechanism underlying this phenomenon

is loss of focal adhesions from G1 to G2 (Fig. 3G–I, Fig. 4; Fig. S3), which is supported by other publications measuring the same outcome (Jones et al., 2018). Previous work has established that focal adhesions control the actin cap confinement of the nucleus (Kim et al., 2012). Both actin and nucleus mechanics change during the cell cycle in a functional manner to move the cell into mitosis, but these changes have real consequences in the integrity of the nucleus and its functions.

## MATERIALS AND METHODS
### Cell culture
MEF WT NLS–GFP ($Lmnb1^{-/-}$ NLS–GFP, $V^{-/-}$) and HT1080 FUCCI cells were cultured in DMEM (Corning) containing 10% fetal bovine serum (FBS, HyClone) and 1% penicillin-streptomycin (Corning). RPE-1 FUCCI were cultured in DMEM/F-12 50/50 (Corning) with 10% FBS and 1% penicillin-streptomycin (complete DMEM). The cells were incubated at 37°C and 5% $CO_2$ with humidity, and passaged every 2–3 days for no more than 30 generations. MEF cells were obtained from the Goldman laboratory (Department of Cell and Molecular Biology, Northwestern University Feinberg School of Medicine, Chicago, IL USA), and HT1080 and RPE-1 FUCCI cells were obtained from Orth laboratory (Department of Molecular and Cellular and Developmental Biology, University of Colorado Boulder, Boulder CO USA).

### Drug treatments
Cells were treated with 4 mM VPA (1069-66-5, Sigma) from a 20 mM dilution in complete DMEM. Cells were treated with 2.5 µM 3-deazaneplanocin (DZNep, 120964-45-6, Cal Biochem) from a 25 mM stock solution in cell culture grade water. Cells were treated with 10 µM RO-3306 (872573-93-8, Sigma) prepared from a 10 mM concentration in DMSO. Cells were treated with 10–25 µM lovostatin (H52792.06, Thermo Fisher Scientific) prepared from a 50 mM stock solution in DMSO. Cells were treated with 5–10 µM FAKi PF-573228 (869288-64-2, Selleckchem) from a 5 mM dilution in DMSO. DZNep, RO-3306 and lovostatin stock solutions did not exceed more than two freeze thaw cycles. Cells were imaged after 16–24 h of treatment with VPA and DZNep, 16 h with RO-3306 and FAKi, and 24 h with lovostatin.

### Immunofluorescence
Cells were plated in eight-well glass chambers (Cellvis) and treated as above. For BrdU experiments, after reaching 70% confluence, cells were treated with 0.03 mg/ml BrdU (B23151, Invitrogen) prepared from a 98 mM stock concentration in DMSO. Cells were treated and incubated with BrdU for 30 min prior to fixation in cold ethanol for 5 min. The cells were then washed three times with PBS (Corning) for 5 min, followed by denaturation with 1.5 M HCl (VWR) for 30 min. For all other immunofluorescence experiments, cells were fixed in a solution of 4% paraformaldehyde in PBS for 15 min and washed three times in PBS for 5 min. Cells were then washed two times with PBS followed by permeabilization with 0.1% Triton X-100 (US Biological) in PBS for 15 min at room temperature, followed by a wash with 0.06% Tween 20 (US Biological) in PBS for 5 min. Cells were washed three times with PBS for 5 min each, followed by blocking for 1 h at room temperature in 2% bovine serum albumin (BSA; 9048-46-8, Fisher) in PBS. Primary antibodies were diluted in 2% BSA and incubated for 12 h at 4°C. Primary antibodies used were: BrdU mouse mAb at 1:1000 (59992s, Cell Signaling Technology), pMLC2 rabbit Ab at 1:100 (3672, Cell Signaling Technology), mouse anti-paxillin at 1:1000 (610052, BD Transduction Laboratories), anti-H2K9ac 1:400 (9649, Cell Signaling Technology), anti-lamin A/C at 1:10,000 (4777, Cell Signaling Technology), anti-H3K9me2-3 1:400 (5327, Cell Signaling Technology) and anti-lamin B1 (1:1000, ab16048 Abcam). Cells were washed three times with PBS. The secondary antibodies used were: Alexa Fluor 555 anti-mouse-IgG (4409S, Cell Signaling Technology), Alexa Fluor 647 anti-mouse-IgG (4410S, Cell Signaling Technology), Alexa Fluor 647 anti-rabbit-IgG (4414, Cell Signaling Technology). Cells were treated with secondary antibodies 1:1000 in blocking solution on the shaker for 1 h at room temperature in the dark. Cells were then washed three times in PBS before staining with a

1 µg/ml dilution of Hoechst 33342 (Life Technologies) in PBS for 15 min followed by three washes in PBS. Cells were mounted using ProLong Gold antifade (Life Technologies) and cured for 12 h at room temperature in the dark. To label RNA, 5-ethynyl uridine (EU) incorporation was detected using the Click-iT RNA Alexa Fluor 594 Imaging Kit (C10330) as previously described (Berg et al., 2023); 1 mM EU was added for 1 h prior to fixation. Following permeabilization, 500 µl of the Click-iT reaction mixture was added and incubated for 30 min in the dark, followed by a rinse in the Click-iT reaction rinse buffer.

### Imaging
Nikon Elements software was used to acquire images on a Nikon instruments Ti2-E inverted widefield microscope, Orca Fusion Gen III camera, Lumencor Celesta light engine, TMC CleanBench air table, with a 40× air objective (NA 0.75, W.D. 0.66, MRH0041), Plan Apochromat Lambda 100× oil immersion objective lens (NA 1.45, W.D. 0.13 mm, MRD71970). For height measurements and imaging of paxillin, a Crest V3 spinning disk confocal was used. Imaging of cell confinement experiments was performed on a Nikon Ti2 inverted widefield microscope using a Plan Apochromat Lambda 20× air objective NA 0.75 (Nikon) with a Prime BSI-Express camera (Photometrics). Nikon Elements software was used for analysis, exported to Excel or Origin, and statistical significance was determined using an unpaired two-tailed Student's $t$-test ($*P<0.05$, $**P<0.01$, $***P<0.001$).

### Immunofluorescence imaging and analysis
Regions of interest (ROIs) were selected to obtain mean intensities of every nuclei. The signal to noise ratio was calculated using mean intensity signal from the ROIs and background taken from every image using a 15×15 pixel square ROI with no cells. A signal to noise ratio greater than 1.5 was the threshold used to identify BrdU positive cells. The cells containing nuclear blebs were compared to the total population by the percent BrdU positive nuclei. To analyze RNA labeling, $Z$-stacks were compiled into a maximum intensity projection and background was subtracted from a 15×15-pixel square region containing no cells. ROIs were drawn around single nuclei and the mean intensities of RNA labeled EU was collected from three fields of view.

Focal adhesion complexes were measured using paxillin immunofluorescence of HT1080 FUCCI cells by confocal imaging with a 100× oil immersion lens in 12-bit sensitive mode and for between 200 and 300 ms at 40–50% power, with focal adhesion complexes were quantified by number of focal adhesions per cell area and sum adhesion area per cell area. Background was subtracted from a 3×3 µm square ROI.

### Live-cell imaging and analysis
Cells were grown on a four-well chamber glass dishes (Cellvis) and treated as above. Cells were treated with 1 µg/ml Hoechst 33342 (Life Technologies) for 15 min prior to live cell imaging. Exposure times for Hoechst 33342 (DAPI), Cdt1–RFP (TRITC) and geminin–GFP (FITC) were between 50 and 200 ms in 12-bit mode. Blebbing was calculated by counting the total number of nuclei with blebs and total nuclei for every replicate. Time lapses were acquired using a humidity chamber complete with Okolab heat and 5% $CO_2$. Images were saved using NIS Elements AR and data was collected in Excel. Nuclear rupture was determined as outlined previously and quantitatively defined nuclear rupture as a >25% increase in the cell-to-nucleus NLS–GFP intensity ratio (Pho et al., 2023).

### Analysis of fluorescent ubiquitin cell cycle indicator
To measure the interphase stage using FUCCI, ROIs were drawn around nuclei and mean intensities of Cdt1–RFP and geminin–GFP were collected using NIS Elements Analysis and exported to Excel. Background from a 15×15-pixel area in a region where there were no cells was collected from every frame to generate signal-to-noise ratios of Cdt1–RFP and geminin–GFP. A signal to noise greater than 1.5 was considered signal, where interphase stages were defined as G1 (Cdt1–RFP positive and geminin–GFP negative), early S (Cdt1–RFP positive and geminin–GFP negative), and late S/G2 (Cdt1–RFP negative and geminin–GFP positive).

Using these thresholds each nucleus was assigned an interphase stage (G1, early S and late S/G2). The percentage population distribution for all

nuclei (total of 100%) for each interphase stage was then graphed. Next each blebbed nucleus was assigned to an interphase stage and it was determined what proportion of those nuclei were in each stage. If the percentage total population of nuclei and percentage of blebbed nuclei for each stage were similar, this suggests population level distributions.

### Live-cell imaging of nuclear height and analysis
Cells were grown in four-well chamber glass dishes (Cellvis) and were untreated or treated 4 mM VPA the day after plating (1069-66-5, Sigma). Cells were stained with 1 µg/ml Hoechst 33342 (Life Technologies) for 15 min prior to imaging. Height images were taken on Nikon Eclipse Ti2 microscope with a 100× oil objective lens with a Crest V3 spinning disk confocal microscope. Images were taken as 77 0.2 µm steps in 12-bit sensitive mode with exposure times for Hoechst 33342 (DAPI), Cdt1–RFP (TRITC) and geminin–GFP (FITC) of between 50 and 100 ms. As described previously (Pho et al., 2023), two intensity line scans were drawn approximately equal in distance from the nucleus center through the Z-plane of each nuclei, values were exported to Excel and the average of the full-width at half-maximum of the two line scans was calculated.

### Live-cell imaging of cell confinement and analysis
To confine the cells, we used a one-well dynamic cell confiner with a suction cup and confiner slides of 3 or 4 µm (4D cell). We hydrated the coverslip and suction cup in complete DMEM for 1 h before use. Confinement was gradually applied by decreasing the pressure from −2 kPa to −10 kPa. Images were taken in FITC and TRITC in 16-bit mode at between 4 and 10% power for 100–200 ms. Loss of compartmentalization of the nucleus was measured as a 20% loss of signal-to-noise of an ROI drawn before and after confinement.

### Micromanipulation force measurements
As previously described (Stephens et al., 2017; Currey et al., 2022), MEF vimentin-null (MEF $V^{-/-}$) were grown in a micromanipulation well. Nikon Ts2R-FL microscope was used to image micromanipulation experiments. Nuclei were isolated from living cells using a spray micropipette containing Triton X-100 (0.05%) in PBS. A pull micropipette was used to grab the nucleus, while the opposite end of the nucleus was grabbed by a precalibrated force micropipette and suspended in preparation for force–extension measurements. The pull pipette was moved 50 nm/s to extend the nucleus 3 or 6 µm. Nucleus extension was measured by tracking the pull micropipette (µm), and force (nN) was measured by deflection of the force micropipette multiplied by the bending modulus (1.2–2 nN/µm). The slope of the force versus extension plot provides the spring constant (nN/µm) for the short chromatin-dominated regime (<3 µm) and long-extension lamin A-dominated strain-stiffening regime (>3 µm). The long-regime spring constant minus the short-regime spring constant provides the measure of lamin A-based strain stiffening. Size thresholds separating interphase stages are reported in Fig. 6.

### Supplemental figures
Fig. S1 provides recapitulation of increased nuclear bleb formation early in the interphase stages (G1) for both MEF and RPE-1 cells. RPE-1 data of changes in nuclear height are also included. Fig. S2 provides data for HT1080 FUCCI and MEF cell lines showing no change in actin contraction as measured by pMLC2 (phorphorylated myosin light chain 2) immunofluorescence throughout the interphase stages. Fig. S3 provides modulation of focal adhesions through FAKi in MEF vimentin null cells which decreases focal adhesions, actin confinement, and nuclear blebbing. Fig. S4 provides data in HT1080 cells showing decrease in H3K9me3 for larger versus smaller nuclei and no change in lamin A/C. Fig. S5 shows that FAKi does not alter loss of H3K9me3 nuclear periphery from G1 to G2 across cell types and conditions.

### Acknowledgements
We would like to thank Dr Kerry Bloom and Dr Pierre Vidi for helpful and insightful discussions. We would like to thank lab members Kelsey Prince and Erin Walsh for helpful discussions. The authors declare no competing interests.

### Competing interests
The authors declare no competing or financial interests.

### Author contributions
Conceptualization: A.D.S.; Data curation: S.B., K.H., A.S.; Formal analysis: S.B., K.H., A.S., S.F., N.L., C.C., N.E., M.P., M.Z.; Investigation: S.B., K.H., A.S., S.F., N.L., C.C., M.Z.; Methodology: K.H., A.S., J.M.; Project administration: A.D.S.; Resources: J.M.; Supervision: L.F.-L., K.B.V., J.O., A.D.S.; Validation: N.E.; Visualization: S.B., K.H., A.S., S.F., M.P., G.M., M.Z.; Writing – original draft: J.O., A.D.S.; Writing – review & editing: L.F.-L., K.B.V.

### Funding
This work was primarily supported by National Institutes of Health (NIH) NIGMS grant Maximizing Investigators' Research Award R35GM154928. This work has also been supported by the Center for 3D Structure and Physics of the Genome 4DN2 grant (1UM1 HG011536). Open Access funding provided through a Read & Publish agreement with University of Massachusetts Amherst. Deposited in PMC for immediate release.

### Data and resource availability
The raw data from all figures are available in a public repository on FigShare (doi:10.6084/m9.figshare.28807298.v2). All other relevant data and details of resources can be found within the article and its supplementary information.

### Peer review history
The peer review history is available online at https://journals.biologists.com/jcs/lookup/doi/10.1242/jcs.264118.reviewer-comments.pdf

### Special Issue
This article is part of the Special Issue 'Cell Biology of the Nucleus', guest edited by Abby Buchwalter. See related articles at https://journals.biologists.com/jcs/issue/139/12.

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
