## [Peer Review File · Journal of Cell Science]

Changes in nuclear and actin mechanics from G1 to G2 affect nuclear integrity

Samantha Bunner, Katie Huang, Anish Shah, Schuyler Figueroa, Nick Lang, Catherine Chu, Nebiyat Eskndir, Mai Pho, Gianna Manning, Mindy Zheng, Lilian Fritz-Laylin, Katrina B. Velle, Joshua Marcus, James Orth and Andrew D. Stephens
DOI: 10.1242/jcs.264118

Editor: Megan King

Review timeline

Original submission:	29 April 2025
Editorial decision:	30 June 2025
First revision received:	22 October 2025
Editorial decision:	24 November 2025
Second revision received:	18 December 2025
Accepted:	8 January 2026

Original submission

First decision letter

MS ID#: jcs.264118

MS TITLE: Changes in nuclear and actin mechanics from G1 to G2 affect nuclear integrity

AUTHORS: Samantha Bunner; Katie Huang; Anish Shah; Nick Lang; Catherine Chu; Schuyler Figueroa; Nebiyat Eskndir; Mai Pho; Gianna Manning; Lilian Fritz-Laylin; Katrina B Velle; Joshua Marcus; James Orth; Andrew D Stephens

ARTICLE TYPE: Research Article

Dear Drew,

We have now reached a decision on the above manuscript.

To see the reviewers' reports and a copy of this decision letter, please go to:

As you will see, the reviewers raise a number of substantial criticisms that prevent me from accepting the paper at this stage. They suggest, however, that a revised version might prove acceptable, if you can address their concerns. If you think that you can deal satisfactorily with the criticisms on revision, I would be pleased to see a revised manuscript. We would then return it to the reviewers. Please pay particular attention to the reviewer comments around the causality of actin-based compression on the nucleus and provide additional experimental data to reinforce your interpretation and/or tone down the conclusions.

Reviewer 1

SUMMARY OF THE ADVANCE MADE IN THIS PAPER AND ITS POTENTIAL SIGNIFICANCE TO THE FIELD

Dr. Stephens' group has published a number of recent papers on nuclear blebbing. This paper presents detailed characterization of the blebbing phenomenon during the cell cycle, which to my

knowledge, has not been reported before. As such, the paper reports novel data. It also reports careful mechanical characterization of the nucleus. Overall, this paper, while not adding significantly more insight into the mechanisms of blebbing itself, is certainly valuable to the field for the careful measurements of the dependence of blebbing on the cell cycle.

SUGGESTIONS TO AUTHORS

1. The conclusion that F-actin confinement is higher in G1 than late S/G2 is really an inference from the observations of a reduction in nuclear height. The height reduction is small - it goes from 6 to 5.2 micron (even if statistically significant, the change is small - 13%). F-actin bundling, or contraction, or some other measurement would be more direct confirmation to prove that the F-actin apical networks are more contractile and cause more confinement of the nucleus. Increased focal adhesion numbers is a promising but indirect measure again. I suggest including some discussion on these nuances to acknowledge limitations of the inference. As an example of an alternative possibility, Dickinson and Lele have shown that excess area in the nuclear lamina can determine nuclear heights in PMID: 37397244. They predict that changing excess area in turn modulates the tension in the nuclear surface, and the pressure difference along the nuclear surface.

2. Some language may need modification- "Finally, dual micropipette micromanipulation single nucleus force measurements confirm that G1 nuclei are stronger than G2 nuclei." The word 'stronger' is ambiguous. Do the authors mean 'stiffer'? There are other sentences on similar lines. e.g. - "Thus, late S/G2 nuclei are weaker than G1 nuclei under similar confinement". Likewise, there is language like "Loss of H3k9me3 causes ... a decrease in chromatin- and lamin-based nuclear mechanics". A 'decrease in nuclear mechanics' is ambiguous, as the word mechanics is not a synonym of the spring constant. Likewise, the title of this paper has the words 'actin mechanics' in it. But actin mechanics is not measured or quantified in anyway in the paper. I recommend toning down the title.

3. The paper does not cite the work of Lele and coworkers in this area, which has sought to explain nuclear dysmorphia in cancer, and also because of their recent proposal of the nuclear drop model, that explains shaping and the mechanism for blebbing.

Reviewer 2

SUMMARY OF THE ADVANCE MADE IN THIS PAPER AND ITS POTENTIAL SIGNIFICANCE TO THE FIELD

The manuscript by Bunner et al. investigates how nuclear and actin mechanics change across the cell cycle and impact nuclear integrity. Using a combination of a FUCCI cell cycle reporter line, quantitative imaging, confinement and micropipette measurements, the authors demonstrate that G1 nuclei are mechanically stiffer than G2 nuclei. They further find differences in heterochromatin levels and focal adhesion organization and propose that these difference influence nuclear rupture in confinement and G2 versus G1 cell cycle stages. This versatile framework is a strength of the study and has the potential to generate useful quantitative data on nuclear mechanics and their role in nuclear bleb formation when the nucleus is deformed by cell-intrinsic forces or cell compression in confined spaces. Overall, the study addresses a timely question on the interplay between cell cycle progression and nuclear mechanics, and their physiological relevance for maintaining nuclear integrity.

SUGGESTIONS TO AUTHORS

While the scope of the study is interesting, the authors make strong claims on causal relations, for example the role of actin and focal adhesions in nuclear ruptures, that are rather correlative and not directly validated. Moreover, several results require clarifications and more details on the involved methodology need to be provided. Also data presentations and statistics require careful revisions for their interpretation. The discussion appears incomplete and the text requires thorough revisions as many spelling errors and typos are present in the manuscript and wordings are often complicated to understand. Given these limitations I do not recommend publication of this current version in JCS.

Major comments:

The role of actin in bleb formation remains unclear. Related to Figure 3D, does LatA treatment reduce bleb formation? The authors show that actin influences cell height, but whether actin directly controls bleb formation as stated by the authors is not explicitly shown.

Similarly, the authors demonstrate that focal adhesions change during interphase stages, but whether focal adhesions directly influence nuclear bleb formation in control versus confined conditions is unclear. Alterations of FA formation or different substrates might be used to address this question.

Figure 4B presents a key conclusion on the susceptibility of G1 nuclei to confinement, but the result is only weakly significant. More replicates might be required and the statistical test used for comparing % values needs to be evaluated for its applicability.

The authors propose that H3K9me3 plays a key role in regulating changes in nuclear mechanics from G1 to G2 stages. They reanalyse perviously published data to support this claim, however it is not shown whether decreasing H3K9me3 levels via chaetocin used in this study also has an influence on nuclear blebbing, which would be relevant for the conclusions of this work. Also, would a decrease in H3K9me3 in G1 cells make them more susceptible to confinement?

The time-lapse data suggest most nuclear blebs originate in G1 and persist through later phases. Do the authors ever observe new bleb formation during S or G2, or are all S/G2 blebs originating from G1 phase?

Several analysis methods used in this study require clarification for their interpretation:

- Fig 1C-E and 3B-C, how was the fraction of nucleus blebs calculated? It is not clear how this analysis was performed. In case % nuclear blebs are provided per specific cell cycle stage it would indicate that G1 cells have a higher rate of nuclear blebs, contrary to the statement of the authors.

- Fig 1C-E, 2C-D and 3B-C, the use of two-tailed Student's t-tests appear incorrect, as the data represent percentages across three mutually exclusive states (which necessarily sum to 100%, causing a correlation between categories). A statistical test should also be used that takes the number of underlying n cells into account. Indeed the differences in data bars is pronounced but mostly no statistically significant differences are indicated as the n numbers are likely not considered, which impacts on the interpretation of the results.

- Lamin KO is not systematically assessed across the different cell types tested and does not seem to be much different than in control embryos

The classification of G1, S, and G2 nuclei is central to the conclusions of the manuscript. To support the robustness of nuclear size thresholds for cell cycle classification, the authors should provide quantitative data on the degree of overlap in nuclear sizes between cell cycle phases.

Various cell types are used in this study for different experiments, making their consistent interpretation difficult. The study would be more compelling if consistent data sets with a single cell line were generated in addition to the provided results - for example analysis nuclear size, blebbing and the role of actin/focal adhesion/H3k9me3 in MEF vimentin null cells used for mechanical measurements of the nucleus.

Minor:

Videos lack time stamps and scale bars

Figure legend 4C and 4D are swapped

Relative measurements are indicated in multiple figures, but it is not indicated how relative measures were obtained. rperiments and methods, making their consistent interpretation difficult.

First revision

Author response to reviewers' comments

Comments from the Reviewers:

Reviewer 1: SUMMARY OF THE ADVANCE MADE IN THIS PAPER AND ITS POTENTIAL SIGNIFICANCE TO THE FIELD

Dr. Stephens' group has published a number of recent papers on nuclear blebbing. This paper presents detailed characterization of the blebbing phenomenon during the cell cycle, which to my knowledge, has not been reported before. As such, the paper reports novel data. It also reports careful mechanical characterization of the nucleus. Overall, this paper, while not adding significantly more insight into the mechanisms of blebbing itself, is certainly valuable to the field for the careful measurements of the dependence of blebbing on the cell cycle.

We appreciate that the reviewer points out that our manuscript is “is certainly valuable to the field for the careful measurements of the dependence of blebbing on the cell cycle”. Furthermore, the reviewer comments that our work “has not been reported before”.

SUGGESTIONS TO AUTHORS

1. The conclusion that F-actin confinement is higher in G1 than late S/G2 is really an inference from the observations of a reduction in nuclear height.

The reviewer raises concerns over sufficient data to support our mechanism. We provide new data showing that the observed changes in focal adhesions can modulate nuclear height/actin confinement and nuclear blebbing (new Figure 4). This new data along with original data provides strong support for our mechanism. Loss of actin confinement via latrunculin A actin depolymerization confirms nuclear height is determined by actin confinement (original data in Figure 3) and affects nuclear blebbing as no new blebs form (new data in Supplemental Figure 1D). This agrees with a long list of published work about the importance of actin confinement in causing nuclear blebbing and deformations (Le Berre et al., 2012; Hatch and Hetzer, 2016; Alabi et al., 2025 MBoC). This also agrees with our work using actin depolymerizer cytochalasin D which decreases actin confinement (Pho et al., 2024 MBoC) and decreases nuclear blebbing (Supplementary Figure Stephens et al., 2018 MBoC). Inducing artificial confinement causes deformations and ruptures as shown in our original data in Figure 5.

The height reduction is small - it goes from 6 to 5.2 micron (even if statistically significant, the change is small - 13%).

We are happy that the reviewer noted this change was statistically significant, which is the agreed upon determination in science. Our new data show that this decrease in height and its major outcome of nuclear blebbing can be reversed with focal adhesion kinase inhibitor (FAKi) in newly added data via new Figure 4 (HT1080) and Supplemental Figure 3 (MEF).

We have previously reported “smaller” changes that were statistically significant in many published papers (Berg et al., 2023 JCS; Pho et al., 2024 MBoC). A 1 μm change under artificial confinement increased nuclear deformation ruptures from < 20% to >80% in our original data in Figure 5.

F-actin bundling, or contraction, or some other measurement would be more direct confirmation to prove that the F-actin apical networks are more contractile and cause more confinement of the nucleus.

In the original manuscript, we provide data showing that actin contraction measured by pMLC2 immunofluorescence does not change throughout the cell cycle across many conditions (Supplemental Figure 2). Many publications support that nuclear blebbing can be modulated independently by only changing actin confinement or actin contraction.

Increased focal adhesion numbers is a promising but indirect measure again. I suggest including some discussion on these nuances to acknowledge limitations of the inference.

New data in the revised manuscript via Figure 4 show that in HT1080 Fucci cells, loss of focal adhesions through focal adhesion kinase inhibition (FAKi) decreases focal adhesions, actin confinement (increased nuclear height), and nuclear blebbing. We also show similar data in MEF cells via new data in new Supplemental Figure 3. As requested by the reviewer, we have added to the discussion a clear connection of our data that is supported by nearly a decade of publications in the field.

As an example of an alternative possibility, Dickinson and Lele have shown that excess area in the nuclear lamina can determine nuclear heights in 37397244. They predict that changing excess area in turn modulates the tension in the nuclear surface, and the pressure difference along the nuclear surface.

The reviewer provides an alternative hypothesis that nuclear height could be due to excess lamins. This hypothesis postulates that G1 has more excess lamins, making it flatter (smaller nuclear height). Going into G2, lamin levels would then decrease, causing the nucleus to be taller. Our data along with many other publications does not support this alternative hypothesis. Lamins are added during S phase as the nucleus grows from G1 to G2. Our immunofluorescence data reveal that lamin A/C levels maintain a similar average amount per unit area (Figure 6H), which refutes this hypothesis. To maintain a similar amount per unit area in a nucleus that is growing in area, more lamins would have to be added. Thus, there is no data supporting loss of lamins from G1 to G2.

Our new data shows that decreased focal adhesions cause increased nuclear height (decreased actin confinement) and decreased nuclear blebbing (Figure 4 and Supplemental Figure 3). Thus, our new data and analysis of our original data show that our hypothesis is supported by the data and that the provided alternative hypothesis is refuted.

2. Some language may need modification- "Finally, dual micropipette micromanipulation single nucleus force measurements confirm that G1 nuclei are stronger than G2 nuclei." The word 'stronger' is ambiguous. Do the authors mean 'stiffer'? There are other sentences on similar lines. e.g. - Thus, late S/G2 nuclei are weaker than G1 nuclei under similar confinement". Likewise, there is language like "Loss of H3k9me3 causes ... a decrease in chromatin- and lamin-based nuclear mechanics". A 'decrease in nuclear mechanics' is ambiguous, as the word mechanics is not a synonym of the spring constant. Likewise, the title of this paper has the words 'actin mechanics' in it. But actin mechanics is not measured or quantified in anyway in the paper. I recommend toning down the title.

In the revised manuscript we have made the suggested changes to clarify wording for the reader. Specifically, we change stronger to stiffer, weaker to more susceptible to rupture, and nuclear mechanics to nuclear spring constant. All changes are highlighted in red text.

The definition of mechanics can mean measurement of forces OR the machinery or working parts of something. Our data, in agreement with a decade's worth of research, supports that actin confinement antagonizes nuclear shape and integrity. In this manuscript we detail the working parts (focal adhesions) that are changing from G1 to G2. We agree that we do not measure actin forces but instead determine the working parts, or mechanics, that alter actin-based nuclear confinement via focal adhesions. Thus, we would like to leave the title unchanged.

3. The paper does not cite the work of Lele and coworkers in this area, which has

sought to explain nuclear dysmorphia in cancer, and also because of their recent proposal of the nuclear drop model, that explains shaping and the mechanism for blebbing.

In the revised manuscript we have included citations of the nuclear drop model.

Reviewer 2: SUMMARY OF THE ADVANCE MADE IN THIS PAPER AND ITS POTENTIAL SIGNIFICANCE TO THE FIELD

The manuscript by Bunner et al. investigates how nuclear and actin mechanics change across the cell cycle and impact nuclear integrity. Using a combination of a FUCCI cell cycle reporter line, quantitative imaging, confinement and micropipette measurements, the authors demonstrate that G1 nuclei are mechanically stiffer than G2 nuclei. They further find differences in heterochromatin levels and focal adhesion organization and propose that these difference influence nuclear rupture in confinement and G2 versus G1 cell cycle stages. This versatile framework is a strength of the study and has the potential to generate useful quantitative data on nuclear mechanics and their role in nuclear bleb formation when the nucleus is deformed by cell-intrinsic forces or cell compression in confined spaces. Overall, the study addresses a timely question on the interplay between cell cycle progression and nuclear mechanics, and their physiological relevance for maintaining nuclear integrity.

We appreciate the reviewer noting that the “versatile framework is a strength of the study” and that the study “study addresses a timely question”.

SUGGESTIONS TO AUTHORS

While the scope of the study is interesting, the authors make strong claims on causal relations, for example the role of actin and focal adhesions in nuclear ruptures, that are rather correlative and not directly validated. Moreover, several results require clarifications and more details on the involved methodology need to be provided. Also data presentations and statistics require careful revisions for their interpretation. The discussion appears incomplete and the text requires thorough revisions as many spelling errors and typos are present in the manuscript and wordings are often complicated to understand. Given these limitations I do not recommend publication of this current version in JCS.

In the revised manuscript we have addressed these points.

1. In the revised manuscript we provide new data modulating focal adhesions using a Focal Adhesion Kinases inhibitor (FAKi) which alters nuclear confinement and nuclear blebbing.
2. We have revised the text to add clarifications of methodology and to clean up the text.

Major comments:

The role of actin in bleb formation remains unclear. Related to Figure 3D, does LatA treatment reduce bleb formation? The authors show that actin influences cell height, but whether actin directly controls bleb formation as stated by the authors is not explicitly shown.

The reviewer is asking for clarification of LatA’s effect on nuclear blebbing. As cited in the original manuscript, many pervious publications have already shown that actin depolymerization results in drastically suppressed nuclear blebbing (Le Berre et al., 2012; Hatch and Hetzer, 2016; Stephens et al., 2018; Cho et al., 2019; Mistriotis et al., 2019; Nmezi et al., 2019; Xia et al., 2019; Pho et al., 2023). Below we share a Supplemental Figure from our previous publication (Stephen et al., 2018 MBoC) showing actin depolymerization of actin via CytoD suppress nuclear blebbing.

This figure has been removed at the authors request. It is supplemental Figure 1E in Stephens AD, Liu PZ, Banigan EJ, Almassalha LM, Backman V, Adam SA, Goldman RD, Marko JF. Chromatin histone

modifications and rigidity affect nuclear morphology independent of lamins. *Mol Biol Cell*. 2018 Jan 15;29(2):220-233. doi: 10.1091/mbc.E17-06-0410. Epub 2017 Nov 15. PMID: 29142071; PMCID: PMC5909933.

To further address this point, we have included data that treatment with LatA suppresses nuclear bleb formation to 0% (Supplemental Figure 1D).

Similarly, the authors demonstrate that focal adhesions change during interphase stages, but whether focal adhesions directly influence nuclear bleb formation in control versus confined conditions is unclear. Alterations of FA formation or different substrates might be used to address this question.

The reviewer suggests that we modify focal adhesions to determine their direct influence on nuclear bleb formation. In the revised manuscript we provide new data in which we treat cells with focal adhesion kinase inhibitor (FAKi). Cells treated with FAKi resulted in decreased focal adhesions, actin confinement, and nuclear blebbing (Figure 4 and Supplemental Figure 3). This data supports that loss of focal adhesions from G1 to G2 shown in Figure 3 is the mechanism underlying these observations.

Figure 4B presents a key conclusion on the susceptibility of G1 nuclei to confinement, but the result is only weakly significant. More replicates might be required and the statistical test used for comparing % values needs to be evaluated for its applicability.

The reviewer requests more biological replicates for original Figure 4 that is now Figure 5 in the revised manuscript. In the revised manuscript we provide two new replicates of 4 um compression, bringing the total to 5 total replicates (Figure 5B, updated). The statistical tests in the original manuscript were statistically significant. Upon addition of the new data the finding remains significant, but we now have a p value of $p < 0.001$.

The authors propose that H3K9me3 plays a key role in regulating changes in nuclear mechanics from G1 to G2 stages. They reanalyse perviously published data to support this claim, however it is not shown whether decreasing H3K9me3 levels via chaetocin used in this study also has an influence on nuclear blebbing, which would be relevant for the conclusions of this work. Also, would a decrease in H3K9me3 in G1 cells make them more susceptible to confinement?

The reviewer is asking if “**decreasing H3K9me3 levels via chaetocin used in this study also has an influence on nuclear blebbing**”. In our recently published work (Manning et al., 2025 Nucleus) we show that chaetocin treatment leads to a two-fold increase in nuclear blebbing for MEF and HT1080 nuclei (Figure 2 of that paper). We have added this clarification to the revised manuscript.

The reviewer is also interested if loss of H3K9me3 from chaetocin (CHT) treatment could make the nucleus more susceptible to actin-based confinement. To address this question, we measured nuclear height in chaetocin treated nuclei and found no statistically significant change ($p = 0.2$, see graph). We felt it unnecessary to add to the manuscript given the lack of a significant change.

The time-lapse data suggest most nuclear blebs originate in G1 and persist through later phases. Do the authors ever observe new bleb formation during S or G2, or are all S/G2 blebs originating from G1 phase?

The reviewer is asking for clarification on nuclear bleb formation during S or G2. In the original data we tracked the formation of all nuclear blebs throughout the cell cycle for 36 hours. In the original manuscript data we find a fraction of blebs do form in S or G2 (30-35%, Figure 3).

Several analysis methods used in this study require clarification for their interpretation:

- Fig 1C-E and 3B-C, how was the fraction of nucleus blebs calculated? It is not clear how this analysis was performed. In case % nuclear blebs are provided per specific cell cycle stage it would indicate that G1 cells have a higher rate of nuclear blebs, contrary to the statement of the authors.

The reviewer is asking for clarification of nuclear blebbing percentages and how they were calculated. For Figure 1C, the legend stated originally that it is the percentage of nuclei with a nuclear bleb in the population. For Figure 1D,E,F the % population distribution is considered, which is different from Figure 1C. Percentage population distribution (total of 100%) for total nuclei in each stage was graphed alongside the blebbed nuclei in each stage. This shows that while more nuclei are in G1 (total) there is not an increase or decrease in blebbed nuclei relative to the total nuclei in any stage G1, early S, late S/G2.

This is a static measurement of cells for Figure 1 which does not account for when the nuclear bleb formed, which is investigated in Figure 3. This same calculation of percentage population distribution was used to track when a nuclear bleb forms using timelapse data in Figure 3 B-C. Time lapse data of FUCCI cells reveals nuclear bleb formation occurs more frequently in G1 nuclei than the number of nuclei in G1. As we mention, since a nuclear bleb forms in G1, on average, it persists to S and G2 providing an explanation to why static measures of nuclear bleb percentages see equal distribution (Figure 1 and 2). We have provided a clarification to the materials and methods section.

- Fig 1C-E, 2C-D and 3B-C, the use of two-tailed Student's t-tests appear incorrect, as the data represent percentages across three mutually exclusive states (which necessarily sum to 100%, causing a correlation between categories). A statistical test should also be used that takes the number of underlying n cells into account. Indeed the differences in data bars is pronounced but mostly no statistically significant differences are indicated as the n numbers are likely not considered, which impacts on the interpretation of the results.

We believe the overall confusion is that we are not comparing across the three mutually exclusive interphase stages (G1, early S, late S/G2). Instead, we are comparing the percentage of the population of nuclei in each interphase stage vs. the percentage of the population of nuclei with a bleb in each interphase stage. Thus, a Student's T-test is used to compare two

conditions against each other. The number of biological replicates is considered, which is 3-4 for the figure panels mentioned and are used for the statistical considerations as each biological replicate provides a percentage. For Figure 1 and 2 the static data reveal no statistically significant differences between total cells and cells with a nuclear bleb for each of the three interphase stages. However, Figure 3 dynamic data reveal a statistically significant difference between total nuclei and nuclei with a bleb formation.

The number of underlying cells that make up each biological replicate is reported in the manuscript. A percentage comes from the whole number of cells per replicate which were similar across conditions.

- Lamin KO is not systematically assessed across the different cell types tested and does not seem to be much different than in control embryos

The reviewer is correct that MEF *Lmn1-/-* is only used in Figure 2. Lamin B1 loss has been reported to elongate S phase. We added it here to compare to DZNep which has a significantly shorter, measured as less of the population in S phase, relative to *Lmn1-/-* (Figure 2C). While these two conditions have different population distributions, the percentage of blebbed nuclei was similar to the population distribution for S phase.

The classification of G1, S, and G2 nuclei is central to the conclusions of the manuscript. To support the robustness of nuclear size thresholds for cell cycle classification, the authors should provide quantitative data on the degree of overlap in nuclear sizes between cell cycle phases.

FUCCI cells are used throughout the paper to report cell cycle stage for HT1080 (Main Figures) and RPE1 (Supplemental Figure 1). Size is used as a proxy for interphase stage for MEF nuclei (Figure 5 and 6 and Supplemental Figures). In the original manuscript we provide cell cycle stalling drugs to verify both the nuclear force measurement data and the size distribution measured in Figure 5B. Using this original data, the nuclear size thresholds in the manuscript are set to minimize possible mixing of G1 into G2 or vice versa to $< \sim 15\%$.

To further address the reviewer's concern, in the revised manuscript, we include new data of Cdt1 immunofluorescence in MEF nuclei to mark cells in G1 (high Cdt1) vs. cells in G2.

Cdt1 immunofluorescence drops off with nuclear size and nuclei greater than $250 \mu\text{m}^2$ show only nominal Cdt signal in $\sim 5\%$ of nuclei (Supplemental Figure 2D).

Figure 5 and 6 measurements are all verified by more than just MEF nuclear size. Nuclear force measurements are taken for both size and for cell cycle stalling drugs. Bulk constitutive heterochromatin and lamin immunofluorescence measurements are done both in MEF based on nuclear size and recapitulated in HT1080 via Supplemental Figure 3. Finally, H3K9me3 peripheral and bulk intensity is also recapitulated in Supplemental Figure 3 as well.

Various cell types are used in this study for different experiments, making their consistent interpretation difficult. The study would be more compelling if consistent data sets with a single cell line were generated in addition to the provided results - for example analysis nuclear size, blebbing and the role of actin/focal adhesion/H3K9me3 in MEF vimentin null cells used for mechanical measurements of the nucleus.

The reviewer questions the use of many cell lines instead of one cell line. Simply put, MEF vimentin null cells provide ease of doing nuclear spring constant measurements via micromanipulation while HT1080 and RPE1 cell lines have live cell markers of the cell cycle via FUCCI for microscopy. We provide a more detailed response below.

1. Overall, many of our published studies use both MEF and HT1080 cells to study nuclear blebbing in two different cell types. We commonly use HT1080 cells alongside MEFs to show cell behaviors are broadly applicable and not cell type dependent. For this study, each cell type had specific benefits listed below.
 - a. HT1080 FUCCI cells provide the ability to track cell cycle, nuclear shape, and

nuclear rupture over time.

- b. Using MEFS, we have a long history of published data on nuclear blebbing, rupture, nuclear height (actin contraction), pMLC2 levels (actin confinement), and many more measurements with this cell type.
 - i. MEF vimentin null cells provide ease of isolation for single nucleus force measurements via micromanipulation. MEF wild type and vimentin KO have similar measured nuclear spring constant (Stephens et al 2017 MBoC; Currey et al. 2022 Cell Mol Bioeng.).
 - ii. MEF wild type and vimentin KO display similar levels of nuclear blebbing for both untreated and past published treatments (Stephens et al., 2018 and 2019 MBoC; Berg et al., 2023 JCS).
 - iii. New data in the revised manuscript shows that MEF vimentin null cells display similar G1 vs. G2 changes in focal adhesions, nuclear confinement and nuclear blebbing, which is dependent on FAK (Supplemental Figure 3). MEF vimentin null nuclei also show similar H3K9me3 periphery changes as shown in MEF WT and HT1080 (Supplemental Figure 5).
- c. RPE1 Fucci cells provide a third cell type to further confirm that the effects are even more broadly applicable (Supplemental Figure 1).

Overall, original and new data in the revised manuscript show that the major outcomes and mechanisms are similar between the two majority used cell lines MEF and HT1080, with an emphasis on showing that MEF vimentin null also provides similar results.

Minor:

Videos lack time stamps and scale bars

We have added scale bars to the timelapses and denote in the legend the total time viewed.

Figure legend 4C and 4D are swapped

We have revised the manuscript to fix this mistake. Thank you for pointing it out.

Relative measurements are indicated in multiple figures, but it is not indicated how relative measures were obtained. rperiments and methods, making their consistent interpretation difficult.

In the revised manuscript we provide an explanation for relative measurements for Figure 1B, Figure 5G and H, Figure 6, and Supplemental Figures.

Second decision letter

MS ID#: jcs.264118R1

MS TITLE: Changes in nuclear and actin mechanics from G1 to G2 affect nuclear integrity

AUTHORS: Samantha Bunner; Katie Huang; Anish Shah; Schuyler Figueroa; Nick Lang; Catherine Chu; Nebiyat Eskndir; Mai Pho; Gianna Manning; Mindy Zheng; Lilian Fritz-Laylin; Katrina B Velle; Joshua Marcus; James Orth; Andrew D Stephens

ARTICLE TYPE: Research Article

Dear Drew,

Thank you again for your revised submission.

The reviewers are generally satisfied with the revised manuscript. I would therefore like to move forward with accepting your manuscript. However, if you could please address the two small suggestions from the reviewers through changes to the text, I would appreciate it.

Reviewer 1

SUMMARY OF THE ADVANCE MADE IN THIS PAPER AND ITS POTENTIAL SIGNIFICANCE TO THE FIELD

The authors have addressed my suggestions.

SUGGESTIONS TO AUTHORS

Minor comments

As a minor comment, I do not agree with the 'refutation' of the excess area mechanism. The word 'excess' in excess area does not mean lamins are produced in excess. It is a geometric concept - every nucleus has excess area in the lamina, which is the area of the nucleus in excess of the area of a sphere of the same volume. Thus, excess area can change without any changes to lamin concentration - simply through changes in volume. And this definitely occurs during cell cycle progression.

Reviewer 2

The authors have sufficiently addressed my comments and included relevant new data, such as the pharmacological inhibition of focal adhesion kinase. Overall, the study presents interesting data that describe changes in nucleus mechanics and bleb formation frequency during the cell cycle, in relation to cell adhesion and the integrity of the actin cytoskeleton.

A minor comment related to the wording in the article: The authors mentioned in their reply to revise the term 'stronger' nuclei, but it is still used throughout the manuscript, abstract etc. I would suggest to remove or clarify this wording explicitly in the article to avoid confusions for the reader.

Second revisionAuthor response to reviewers' comments

Comments from the Reviewers:

Reviewer 1: SUMMARY OF THE ADVANCE MADE IN THIS PAPER AND ITS POTENTIAL SIGNIFICANCE TO THE FIELD

The authors have addressed my suggestions.

SUGGESTIONS TO AUTHORS

Reviewer 1: Minor comments

As a minor comment, I do not agree with the 'refutation' of the excess area mechanism. The word 'excess' in excess area does not mean lamins are produced in excess. It is a geometric concept - every nucleus has excess area in the lamina, which is the area of the nucleus in excess of the area of a sphere of the same volume. Thus, excess area can change without any changes to lamin concentration - simply through changes in volume. And this definitely occurs during cell cycle progression.

Authors: We appreciate the reviewer's feedback. We agree that excess area does not mean lamins are produced in excess. This comment was only in the review and not the manuscript.

Reviewer 2: The authors have sufficiently addressed my comments and included relevant new data, such as the pharmacological inhibition of focal adhesion kinase. Overall, the study presents interesting data that describe changes in nucleus mechanics and bleb formation frequency during the cell cycle, in relation to cell adhesion and the integrity of

the actin cytoskeleton.

Authors: We appreciate the reviewers' feedback and are pleased to hear that the reviewer states our manuscript presents interesting data.

Reviewer 2: A minor comment related to the wording in the article: The authors mentioned in their reply to revise the term 'stronger' nuclei, but it is still used throughout the manuscript, abstract etc. I would suggest to remove or clarify this wording explicitly in the article to avoid confusions for the reader.

Authors: We appreciate the reviewers' feedback and catching this word. We have now replaced stronger with stiffer and strong with rigid.

Third decision letter

MS ID#: jcs.264118R2

MS Title: Changes in nuclear and actin mechanics from G1 to G2 affect nuclear integrity

Authors: Samantha Bunner; Katie Huang; Anish Shah; Schuyler Figueroa; Nick Lang; Catherine Chu; Nebiyat Eskndir; Mai Pho; Gianna Manning; Mindy Zheng; Lilian Fritz-Laylin; Katrina B Velle; Joshua Marcus; James Orth; Andrew D Stephens

Article Type: Research Article

Dear Drew,

Happy New Year. Thank you for addressing the two small comments. I am happy to tell you that your manuscript has been accepted for publication in Journal of Cell Science, pending standard publication integrity checks.